# When do World Models Successfully Learn Dynamical Systems?

## Abstract

In this work, we explore the use of compact latent representations with learned time dynamics ('World Models') to simulate physical systems. Drawing on concepts from control theory, we propose a theoretical framework that explains why projecting time slices into a low-dimensional space and then concatenating to form a history ('Tokenization') is effective at learning dynamical systems, and characterise when the underlying dynamics admit a reconstruction mapping from the history of previous tokenized frames to the next, with the key novel insight being observability. To validate these claims, we develop a sequence of models with increasing complexity, starting with least-squares regression and progressing through simple linear layers, shallow adversarial learners, and ultimately full-scale generative adversarial networks (GANs). We evaluate these models on a variety of datasets, including modified forms of the heat and wave equations, the chaotic regime 2D Kuramoto-Sivashinsky equation, and a challenging computational fluid dynamics (CFD) dataset of a 2D Kármán vortex street around a fixed cylinder, where our model is successfully able to recreate the flow. Comparisons to FNO and DeepONet show comparable short- and improved long-term accuracy.

## 1 Introduction

World models were introduced by Ha et. al. (Ha & Schmidhuber, 2018) as generative models that compress high-dimensional states into latent variables with learned autoregressive dynamics. Lecun (LeCun, 2022) emphasises that latent dynamics should encode the physical laws governing the environment rather than reproducing observational correlations. This connects world models to state-space modelling, observability, and operator learning. Recent advances follow this paradigm by using powerful autoregressive sequence models to capture temporal structure in the latent space, combined with super-resolution modules that reconstruct high-resolution states. This approach has produced stunning results in video generation and dynamic scene synthesis (Hu et al., 2023; Peng et al., 2025).

Rather than as generative models, world models can be understood more fundamentally as learned dynamical systems. The encoder acts as an observation operator that compresses a short temporal window of the full state into a low-dimensional representation, while the latent dynamics module predicts the next latent state based on this compressed history. A reconstruction module then lifts the latent representation back to the full-dimensional state. Interpreted system-theoretically, this architecture defines a learned reduced-order state-space model. Its effectiveness depends on whether the compressed representation preserves information necessary for identifying the underlying physical state.

This system-theoretic view also connects conceptually to neural operator learning, where the evolution of physical systems is learned from data governed by partial differential equations (Li et al., 2021; Lu et al., 2021; Kovachki et al., 2023; Zhang et al., 2025). Neural operators are trained on sample trajectories produced by numerical simulation and then generalise to new initial conditions, from which full solutions to the physical equations can be recovered.

Both world models and neural operators aim to approximate the underlying dynamical operator, but most autoregressive operators remain one-step predictors. We therefore study how latent-history length relates to the system's effective order and how observability is preserved under compression.

Generative modelling of physical systems has already been widely explored with an emphasis on turbulent flows, see e.g. (Kim & Lee, 2020; Drygala et al., 2022). A few works have also explored spatio-temporal generative modelling of the type discussed in Section 1. However, these works rely primarily on video similarity metrics and do not provide theoretical or empirical studies of the physical accuracy of the generated data. World models have also been applied to simulate physical systems (Skorokhodov et al., 2022; Klemmer et al., 2022), but these efforts focus on video generation and therefore neither analyse the system-theoretic foundations nor compare their performance to operator-learning approaches, leaving the research questions stated above open.

Our contributions in this paper are as follows:

(1) We introduce a system-theoretic framework to understand when world models can reliably learn dynamics from tokenized observations. The study is mathematically grounded, interpreting world-models within operator learning, and shows that observability is a key ingredient. Video visualisations of our solutions can be found in the supplementary material.

(2) We compare this approach to state of the art methods like FNO and DeepONet, showing superior long-term stability of world models for in- and out-of-distribution test data.

(3) We provide numerical studies on a hierarchy of dynamical systems on the length of token history in autoregressive predictions and super-resolution reconstruction, which is a first study of autoregressive model order reduction for dynamical systems. Models perform significantly worse when lacking the observability assumption.

(4) We show that world-model based simulation can achieve low error predictions in capturing the key temporal correlations in the case of turbulence modelling even in partial information situations, at significantly less computational cost than traditional methods from computational fluid dynamics.

## 2 Related Work

**Generative super-resolution reconstruction of turbulent flow**  A prominent application of generative learning in turbulence is resolution reconstruction. GAN-based methods dominate this area, with (Enhanced) Super-Resolution GAN (Wang et al., 2018) variants widely used to recover high-resolution turbulent fields from low-resolution or noisy inputs, often enhanced by physics-informed loss functions (Deng et al., 2019; Bode et al., 2023; Chen et al., 2024; Zheng et al., 2024). More recently, (Zhang et al., 2025) proposed an operator-learning approach that reconstructs high-resolution fields from sparse data using a tokenizer and energy transformer.

**Spatio-temporal generative modelling for turbulent flow.**  Moving beyond snapshot reconstruction, recent studies have applied generative learning to spatio-temporal turbulence modelling. Transformer-based frameworks encode temporal interactions into latent spaces, guided by Navier–Stokes constraints (Xu et al., 2025) or spatial embeddings (Li et al., 2022; Alkin et al., 2024). Recurrent neural networks integrate physics through boundary conditions (Ren et al., 2023; 2025) or pretrained modules (Wan et al., 2025). Residual networks (Liu et al., 2024) and diffusion models (Rühling Cachay et al., 2023) have also been applied, with diffusion models notable for operating in a purely data-driven manner (Kohl et al., 2024; Yang et al., 2023), but limited by slower inference. Other directions include autoregressive pretraining for surrogates (McCabe et al., 2024), GAN–Autoencoder hybrids for next-step prediction (Afzali et al., 2021), and world models adapted from video generation (Skorokhodov et al., 2022; Klemmer et al., 2022).

**Operator learning** focuses on approximating mappings between infinite-dimensional function spaces, allowing models to capture entire solution operators rather than just pointwise predictions. Approaches such as Physics-informed Neural Networks (PINN) (Raissi et al., 2019) and Neural ODE (Chen et al., 2018) utilise neural networks to solve specific ordinary or partial differential equations (ODE/PDE), either by embedding physical laws into the loss function or by parametrising continuous-time dynamics. These methods are generalised by neural operator learning, which aims to efficiently generate solutions for entire families of PDEs by instead learning solution operators, which take as input both the problem specification (e.g. coefficients, forcing terms, boundary or initial conditions) and the associated function space, and output the corresponding solution function. Within this framework, Fourier Neural Operators (FNO) (Li et al., 2021; Kovachki et al., 2023) perform convolutions in the spectral domain to capture long-range interactions, while DeepONets (Lu et al., 2021) employ a branch-trunk architecture to flexibly map input functions to outputs. For a comprehensive overview of (neural) operator learning methods and frameworks, see Cai et al. (2021); Tanyu et al. (2023); Kovachki et al. (2024)

## 3 Theoretical Results

### 3.1 Dynamical systems, tokenization and autoregressive dynamics

In this section we collect some facts from system theory for the mathematical understanding of world models. We consider a dynamical system

$$\dot{x}(t) = f\big(x(t)\big), \qquad x(0) = x_0 \in \mathcal{X} \subset \mathbb{R}^n, \tag{1}$$

where the state space $\mathcal{X}$ is compact with differentiable boundary and $f \colon \mathbb{R}^n \to \mathbb{R}^n$ is Lipschitz with constant $L > 0$ and fulfils $f(x) \cdot \nu(x) = 0$ for $x \in \partial\mathcal{X}$, where $\partial\mathcal{X}$ is the boundary of $\mathcal{X}$ and $\nu(x)$ the outward normal vector field. The Picard–Lindelöf theorem (Teschl, 2012) ensures global existence and uniqueness of solutions. Furthermore, as $f(x)$ is tangential to $\mathcal{X}$ at the boundary, the solution never leaves $\mathcal{X}$. If $x(t)$ and $\tilde{x}(t)$ start from $x_0$ and $\tilde{x}_0$, the Grönwall estimate implies continuous dependence on initial data.

Let $h \colon \mathcal{X} \to \mathbb{R}^m$ $(m \leq n)$ be a continuous tokenization map which maps the high dimensional state $x \in \mathcal{X}$ into the latent representation $y = h(x) \in \mathbb{R}^m$ with $m \ll n$. In system theory $h$ is known as the (system) output map. We define the tokenized (latent) dynamics

$$y(t) := h\big(x(t)\big) \in \mathbb{R}^m. \tag{2}$$

Denote by $S_\Delta \colon \mathcal{X} \to \mathcal{X}$ the time–$\Delta$ flow ("propagator") associated with (1):

$$S_\Delta(x) := x(\Delta) \quad \text{when } x(0) = x. \tag{3}$$

For a fixed integer $k \geq 1$ we collect $k$ equidistant past outputs into the *measurement sequence*

$$y_{\mathrm{seq}}(t) := \big(y(t - (k-1)\Delta), \, y(t - (k-2)\Delta), \, \ldots, \, y(t)\big) \in \mathcal{Y}^k := \big(h(\mathcal{X})\big)^k. \tag{4}$$

The following definition introduces the system theoretic and the world model perspectives on the tokenized dynamics (1) and (4).

**Definition 1.** *(i) Given $x_0 \in \mathcal{X}$ and $\Delta, T > 0$ such that $\frac{T}{\Delta} \in \mathbb{N}$. A state $\tilde{x}_0$ is* indistinguish-able *(on $\Gamma_T = \{\Delta j : j \in \{0, 1, \ldots, \frac{T}{\Delta}\}\}$) from $x_0$ if the associated outputs coincide, i.e., if it holds $y(t; x_0) \equiv y(t; \tilde{x}_0)$ for $t \in \Gamma_T$. The set of states which are indistinguishable from $x_0$ is denoted by $\mathcal{I}(x_0)$. The pair $(f, h)$ is* observable *if for any $x_0 \in \mathcal{X}$ there exists $\Delta, T > 0$ s.t. $\mathcal{I}(x_0) = \{x_0\}$. The pair $(f, h)$ is* observable *if $(f, h)$ is observable for all $x_0 \in \mathcal{S}$. Hermann & Krener (1977) on $\mathcal{S}$, if there exists $k \in \mathbb{N}$ and a continuous reconstruction map*

$$G \colon \big(h(\mathcal{S})\big)^k \longrightarrow \mathcal{S}, \qquad G\big(y_{\mathrm{seq}}(t)\big) = x(t). \tag{5}$$

*(ii) An autoregressive dynamics on the latent space $\mathcal{Y}$ with history length $k \in \mathbb{N}$ and time step $\Delta > 0$ is a mapping*

$$g \colon \mathcal{Y}^k \to \mathcal{Y} \quad \text{such that} \quad g(y_{\mathrm{seq}}(t)) = y(t + \Delta). \tag{6}$$

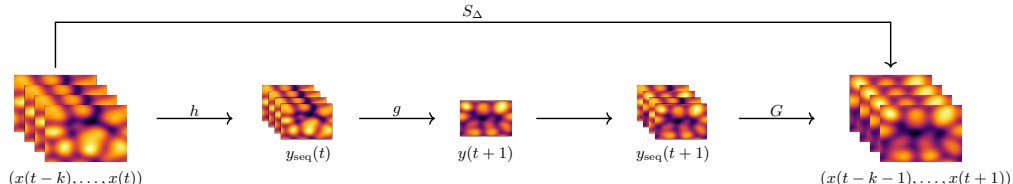

Figure 1: The full-state time-series data $x_{t-k:t} = (x(t - k), \ldots, x(t))$ is tokenized via $y_{\text{seq}}(t) = h(x_{t-k:t})$, yielding a low-dimensional representation. The next observation is predicted by $y(t+1) = g(y_{\text{seq}}(t))$, then appended—dropping the oldest entry— to form $y_{\text{seq}}(t+1)$, and finally the full state is reconstructed via $x(t+1) = G(y_{\text{seq}}(t+1))$.

Observability concepts for nonlinear control systems have been rigorously analysed in, e.g., Hermann & Krener (1977). Further details on observability in the standard, continuous setting of control theory can be found in Appendix B.

In the context of world models, the reconstruction map $G$ is also called a decoder or super-resolution map (Fig. 1). Observability (i) and (ii) are closely related, as the following Lemma shows:

**Lemma 2.** *Suppose the dynamical system is observable in the sense of Definition 1 (i). Then the autoregressive dynamics defined in Definition 1 (ii) exists.*

*Proof.* Composing $G$ with the flow and measurement functions yields $g \colon \mathcal{Y}^k \longrightarrow \mathcal{Y}$ via $g := h \circ S_\Delta \circ G$. As $G$, $S_\Delta$ and $h$ are continuous, so is $g$. In fact,

$$g\big(y_{\text{seq}}(t)\big) = h \circ S_\Delta \circ G\big(y_{\text{seq}}(t)\big) = h(S_\Delta(x(t))) = h(x(t+\Delta)) = y(t+\Delta). \qquad \square$$

System theory gives us sufficient conditions for the existence of a reconstruction map $G$ and hence an autoregressive dynamics $g$ in the latent space.

In our experiments we distinguish two cases: linear observability and nonlinear observability.

A linear dynamical system is given by $S_\Delta(x) = e^{\Delta A} x$ where $A \in \mathbb{R}^{n \times n}$ is the generator of the dynamics, i.e. in (1) we have $f(x) = Ax$. In this easy case, we can drop the compactness condition for the state space $\mathcal{X}$ and identify it with $\mathbb{R}^n$. If we additionally have a linear observation $h \colon \mathbb{R}^n \to \mathbb{R}^m, h(x) = hx$ with $h \in \mathbb{R}^{m \times n}$, observability is well-known to be equivalently characterized in terms of algebraic rank conditions (Sontag, 1998, Chapter 6). In particular, the seminal work of Kalman (1963) related observability to the (Kalman) observability matrix which for $k \in \mathbb{N}$ is defined as

$$\mathcal{O} = \begin{pmatrix} h^\top & (hA)^\top & \cdots & (hA^{n-1})^\top \end{pmatrix}^\top \in \mathbb{R}^{nm \times n}. \tag{7}$$

Then the following theorem characterizes linear observability.

**Theorem 3.** *The following are equivalent. (i) The linear dynamical system is linearly observable; (ii) The observability matrix has full rank $n$; (iii) For all eigenvectors $v$ of $A$ we have $hv \neq 0$.*

*Proof.* For the proof of this standard result see, e.g., (Sontag, 1998, Chapter 3). Note that the third condition is typically referred to as the (Popov-Belevitch-)Hautus test for observability. $\square$

Unfortunately, the situation in the non-linear case is less well understood. We refer to Appendix B for a detailed discussion of the state of research.

### 3.2 Operator learning using the world model approach

On the basis of the assumption of global observability through a reconstruction map $G \colon \mathcal{Y}^k \to \mathcal{X}$, we develop the statistical learning theory for the autoregressive map $g$, the reconstruction

map $G$ and the dynamics operator $S_\Delta$. As $G$ is only assumed to be continuous, we refrain to qualitative results of PAC learnability and do not aim to give rates of convergence.

The training data is given by a set of $N$ trajectories $x(t; x_0^{(j)})$ with initial conditions $x(0; x_0^{(j)}) = x_0^{(j)}$, $j = 1, \ldots, N$, where $x_0^{(j)} \sim \mu$ are drawn independently from some distribution $\mu$ on $\mathcal{X}$ with continuous density function $d\mu(x) = f_\mu(x)dx$ such that $f_\mu(x) > 0$ on $\mathcal{X}$. The training data is observed at times $t \in \Gamma_T = \{\Delta l : k \in \mathbb{N}, \Delta l \leq T\}$ so that we have $\frac{T}{\Delta} + 1$ time steps. Applying the tokenization map $h$, we furthermore have access to $y(t; x_0^{(j)}) = h(x(t; x_0^j))$, $t \in \Gamma_t$. Suppose that $k$ is sufficiently high such that histories of length $k$, $y_{\text{seq}}(t; x_0) = (y(t - (k-1)\Delta; x_0), y(t - (k-1)\Delta; x_0), \ldots, y(t; x_0))$, $t \in \Gamma_T$ suffice to reconstruct $x(t; x_0) = G(y_{\text{seq}}^{(X)}(t))$. We consider the risk functions for $g$ and $G$ given by

$$\mathcal{L}_{g,T}(\eta_g) = \mathbb{E}_{x_0 \sim \mu, t \sim U(\Gamma_T \setminus (\Gamma_{\Delta(k-1)} \cup \{T\}))} \left[ \|\eta_g(y_{\text{seq}}(t; x_0)) - y(t + \Delta; x_0)\|^2 \right], \quad \text{(8a)}$$

$$\mathcal{L}_{G,T}(\eta_G^\theta) = \mathbb{E}_{x_0 \sim \mu, t \sim U(\Gamma_T \setminus \Gamma_{\Delta(k-1)})} \left[ \|\eta_G(y_{\text{seq}}(t; x_0)) - x(t; x_0)\|^2 \right], \quad \text{(8b)}$$

where $\eta_g : \mathcal{Y}^{\times k} \to \mathcal{Y}$ and $\eta_G : \mathcal{Y}^{\times k} \to \mathcal{X}$ are represented by neural networks from hypothesis spaces $\mathcal{H}_{g,N,T}$ and $\mathcal{H}_{G,N,T}$, respectively and $U(A)$ stands for the uniform distribution on the finite set $A$. Let $\hat{\mathcal{L}}_{N,T,g}(\eta_g)$ and $\hat{\mathcal{L}}_{N,T,G}(\eta_G)$ be the corresponding empirical risk function learning from $N$ trajectories of length $T$, and $\hat{\eta}_{g,N,T}$, $\hat{\eta}_G$ where they take their respective minimums. See Appendix A for definitions.

We obtain the following theorem on probably approximately correct (PAC) learning, see Appendix A for the proofs in this section.

**Theorem 4** (PAC-learning of $g$ and $G$). *Let $\varepsilon, \delta > 0$ be arbitrary. Then there exist hypothesis spaces $\mathcal{H}_{g,N,T}$, $\mathcal{H}_{G,N,T}$ of neural networks and a function $N(\varepsilon, \delta)$ such that*

$$P(\mathcal{L}_{g,N(\varepsilon,\delta),T}(\hat{\eta}_{g,N(\varepsilon,\delta),T}) \leq \varepsilon) \geq 1 - \delta \text{ and } P(\mathcal{L}_{G,N(\varepsilon,\delta),T}(\hat{\eta}_{G,N(\varepsilon,\delta),T}) \leq \varepsilon) \geq 1 - \delta \quad \text{(9)}$$

Let us next consider a random state $x_0 \sim \mu$ and its past token sequence $y_{\text{past}}(x_0)$ of length $k$ and $y_{\text{past}-}(x_0)$ on length $(k-1)$. We define the risk function measuring the error in the autoregressive update of the full state $S_\Delta(X)$ as follows

$$\mathcal{L}_S(\eta_g, \eta_G) = \mathbb{E}_{x_0 \sim \mu} \left[ \|S_\Delta(x_0) - \eta_G(y_{\text{past}-}(x_0), \eta_g(y_{\text{past}}(x_0))\|^2 \right]. \quad \text{(10)}$$

We then get:

**Theorem 5** (Autoregressive PAC-Learning of $S_\Delta$). *Let $\epsilon, \delta > 0$, then there exits $\varepsilon', \delta' > 0$ such that*

$$P(\mathcal{L}_S(\hat{\eta}_{g,N(\varepsilon',\delta'),T}, \hat{\eta}_{G,N(\varepsilon',\delta'),T}) \leq \varepsilon) \geq 1 - \delta.$$

The autoregressive approach can also be trained variationally in GAN-fashion, see Appendix D for the details.

### 3.3 EXAMPLES

The examples we consider are spatially or spatio-temporally discretized partial differential equations (PDE) on a quadratic lattice, see Appendix E. In all cases we utilize a simple tokenizer map $h(x)$ used e.g. in Brooks et al. (2022), taking the average on patches of pixel size $\sqrt{\kappa} \times \sqrt{\kappa}$ with $\sqrt{\kappa} \in \mathbb{N}$ and $m = n/\kappa \in \mathbb{N}$.

**Linear equations.** Let $a(\xi) \geq 0$, $\xi \in \Gamma$, be a thermal conductivity coefficient, which is constant in $t$ but not necessarily constant in space. The heat and wave equation, respectively, is given by

$$\frac{\partial u}{\partial t} = \nabla \cdot (a \nabla u), \quad \frac{\partial^2 u}{\partial t^2} = \nabla \cdot (a \nabla u), \quad \text{(11)}$$

In the heat equation, the state $x(t)$ in equation (1) then consists of the pixel values $x(t) = (u(t, \xi))_{\xi \in \Gamma}$ and the dimension is $n = |\Gamma|$, the number of points in $\Gamma$. The linear dynamics then can be written as $\dot{x}(t) = A_h x(t)$ with $A_h = \nabla \cdot (a \nabla)$ the lattice Laplace operator with conductivity $a$.

In the latter wave equation, $\sqrt{a(\xi)} > 0$ stands for the local wave propagation speed. As the wave equation is second order, the state of the dynamical system $x(t) = (u(t, \xi), v(t, \xi))_{\xi \in \Gamma}$ not only contains the wave amplitudes $u(t, \xi)$ but also the momentum degrees of freedom $v(t, \xi) = \frac{\partial}{\partial t} u(t, \xi)$ and we have $n = 2|\Gamma|$ as state dimension. the dynamical system can then be written in a first order formulation as follows:

$$\dot{x}(t) = \frac{\partial}{\partial t} \begin{pmatrix} u \\ v \end{pmatrix} = \left( \begin{array}{c|c} 0 & \mathbb{1} \\ \hline \nabla \cdot (a\nabla) & 0 \end{array} \right) \begin{pmatrix} u \\ v \end{pmatrix} = A_w x(t). \tag{12}$$

The token map $h$ employed averages wave amplitudes $u(t, \xi)$ over squared regions containing $k$ pixels $\xi$.

The following theorem provides a warning that situations exist where world models will generally fail to learn the dynamical system due to the lack of observability.

**Theorem 6.** *Let the coefficient function $a = a(\xi)$ be constant for the heat and wave equation. Then the respective dynamical system is not observable.*

See Appendix A for the proof. Our experiments in Fig. 2 show a significant deterioration of operator learning with world models for the case of the heat equation with constant coefficients. To produce positive results, we thus perturb the lattice Laplacian using a non constant eigenfunction $a(\xi) = \exp(Z(\xi))$, where $Z(\xi)$ is a restriction of a realization of a Gaussian random field to $\Gamma$. Numerical tests based on diagonalization of the so-obtained matrix show that generally the eigenvectors are not annihilated by the tokenizer $h$ for non constant coefficients. Correspondingly, numerical errors are much reduced.

**Nonlinear equations.** The Kuramoto–Sivashinsky equation (KSE) is considered one of the simplest PDEs with chaotic behavior and is given by

$$\frac{\partial u}{\partial t} + u\nabla u + \nabla^2 u + (\nabla^2)^2 u = 0, \tag{13}$$

where $\nabla^2$ is the Laplace operator $\nabla \cdot \nabla$. Observability of the KSE is discussed in Appendix F.

As a second experiment with non-linear dynamics, we consider the flow around a circular cylinder at a Reynolds number of 3900 presents a complex fluid dynamic behaviour since the flow exhibits characteristics of both laminar and turbulent regimes, with vortex shedding forming a Kármán vortex street downstream of the cylinder. The viscous fluid is governed by the Navier Stokes equations (NSE), which stem from the conservation laws of mass, momentum, and energy where the mass simplifies to the divergence-free nature of the velocity field. Moreover the energy equation can be neglected since the flow in the present scenario is incompressible and isothermal. Given the velocity vector $\mathbf{u} = (u, v, z)$ we can write the NSE in Cartesian coordinates as

$$\nabla \cdot \mathbf{u} = 0, \quad \frac{\partial \mathbf{u}}{\partial t} + (\mathbf{u} \cdot \nabla)\mathbf{u} = -\frac{1}{\rho}\nabla p + \nu\nabla^2\mathbf{u}, \tag{14}$$

where $\rho$ is the fluid density, $\nu$ is the kinematic viscosity, and $p$ denotes the pressure field.

Following Drygala et al. (2022) Eq. (24), in this experiment, we use a non-linear tokenization which only observes the turbulence strength, which measures the deviation of the absolute velocity from the long time average. This gives a scalar turbulence field, to which thereafter the standard tokenizer of LongVideoGAN (Brooks et al., 2022) is applied. The reconstruction map $G$ is not considered for the full state, but for the full state projected to the turbulence strength, which is a continuous mapping. Observability of this projected state follows from the observability of the full state, which is unknown. The quality of the world model reconstructions thus has to serve as experimental evidence of observability.

## 4 Datasets and numerical settings

In order to verify the theoretical results of Section 3, we created datasets for the heat, wave, and KS equations. To verify the methodology on a standard computational fluid dynamics

problem, we also included a published dataset for flow around a cylinder at Reynolds number 3900, a well-studied case in the literature. More details are given in Section G. The datasets are learned with a variety of methods, starting with a simple least squares regression model, to gradient descent methods, through to full DCGAN methods employed by NVIDIA's LongVideoGAN (Brooks et al., 2022).

It is important to point out that all of the datasets used in this work are fundamentally *dynamical*, i.e. they are videos. We give some static snapshots (Figs. 7, 8, 9, 17) but in order to get a proper impression of the both the dataset and model outputs, the reader is strongly encouraged to examine the videos in the supplementary material.

The heat, wave and KS datasets use a Gaussian random field as the initial condition of the flow field $u$. Where applicable, the initial velocity field $v$ was identically set to 0. All three datasets use periodic boundary conditions. The precise numerical parameters used to generate the Gaussian random field are listed as Table 10 in Appendix G.

## 5 EMPIRICAL RESULTS

### 5.1 LINEAR EQUATIONS

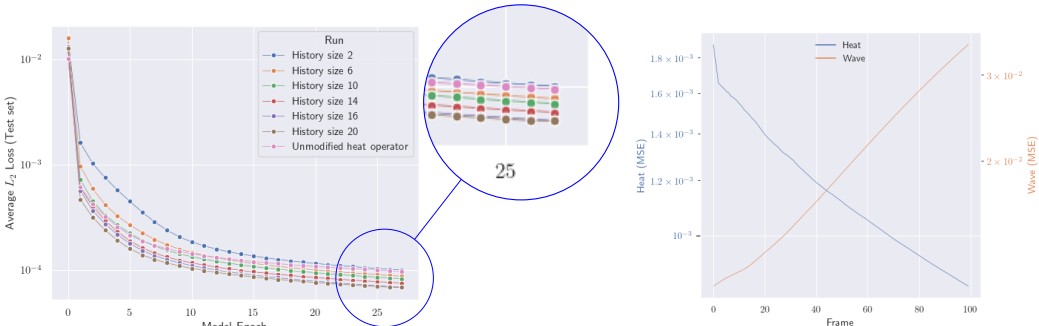

Figure 2: Average $L_2$ residues of our low-res heat equation model against epoch, for the test set. Each x-axis unit represents 2000 epochs. Data is normalised between $[-1, 1]$. The unmodified (non-observable) heat equation has history size 16.

Figure 3: Full model $L_2$ error results, for the heat and wave equations. Each low-res model is fed the first 16 frames from the test set, and is left to generate the next 100 using its own output. The result is then fed to the super-res model.

We report results obtained with our approach: an autoregressive model using a large history in low dimension, whose output is passed to a super-resolution autoregressive model, rather than relying solely on the previous frame as is common in dynamical-systems models. We demonstrate this approach by using both modern gradient-based methods and traditional multivariate linear regression to validate the machine-learning techniques. Thm. 7 ensures the reconstruction map exists under observability; it remains to check it can be found in practice.

**Least-Squares Regression.** We begin in a fully specified statistical model for the low resolution model, where the learning is done by multivariate least squares regression. Fig.15 shows the average absolute error of the residues when the history length of the model is varied, whilst Fig. 16 shows the maximum error across all pixels in all 2000 frames. Each point is sampled 100 times. The shaded area represents a $1\sigma$ range. Note the significant drop in error; we are guaranteed by Lemma 2 that this occurs as the supplied history length to the model exceeds the compression factor $k$. We can see that in this 'ideal' case (optimality given by Gauss-Markov theorem) the drop actually happens much earlier.

**Autoregressive Learning.** Fig. 2 shows how the error evolves against steps for stochastic gradient descent via the Adam (Kingma & Ba, 2017) optimiser. The model shows a clear improvement as history size increases, but past the compression factor of 16, increasing the

history size provides no benefit. In particular, the runs with history size 16 and 20 have converged by the end of training. Note also that the model performs poorly for an unmodified (non-observable) heat operator. This demonstrates that observability is a necessary condition for learning, as stated in Section 3, and adding more information beyond observability doesn't change the end result.

**Full Pipeline Error.** The same analysis can be carried out for the full approach. Fig. 3 shows the full approach error for 100 frames of generated video. As expected, the error increases with time for the wave equation, but in fact decreases in time for the heat equation, due to both the model and the ground truth decaying to the low entropy state.

### 5.2 Non-linear equations

Loss curves for frame-wise $L_1$ and $L_2$ residue loss can be found in Fig. 13. We computed $\rho(\Delta t)$ (See Section H.3) for 15 generated and 15 real 300 frame videos from the test set. We compare the sample means and standard deviation in Fig. 14. Note the model has learned this statistic well for early times, but appears to 'desync' at longer time horizons.

**Comparison to Existing Models.** A direct comparison of our method with Li et al. (2021) and Lu et al. (2021) can be found in Table 1. Both were trained in a data-driven manner; the DeepONet uses an FNO as the branch network and an FCN as the trunk. We used the FNO architecture from NVIDIA's PhysicsNeMo (Contributors, 2023), and a self built DeepONet. For the linear equations, we compute the MSE and L1 (error) norm for a 1, 200, and 1000 frame look-ahead prediction. Initial conditions were sampled random fields distinct from the training data, but with the same distribution. For the out-of-distribution case, see Table 5. The KSE includes a 1 frame look-ahead MSE and L1 norm, as well as a metric measuring the distance between the correograms (x-axis 1-200 frames) of the model and ground truth data (Table 4). Here, 'diverged' refers to a value larger than float precision. We note that our model obtains better performance metrics in the majority of cases. This is especially true of long time horizons, which underlines the stabilising effect of our approach. Further comparisons, a small ablation study and exact implementation details can be found in Section C.

We omit a comparison to PINNs (Raissi et al., 2019), as PINNs need to be retrained for new initial conditions, which affects robustness and speed at generation time. Additionally, the KVS dataset contains partial information of the state, and thus the data driven case is more general. Current experimental methods (e.g. Ghazijahani & Cierpka (2024)) also only give a partial measurement. The theoretical guarantees given by Theorems 4 and 5 make sure that despite this, assuming observability, the physics is learned accurately.

### 5.3 Flow around a cylinder.

We computed $\rho(\Delta t)$ for $\Delta t \in \{1, \ldots, 100\}$ and for 4 different choices of $P$ for 100 non-overlapping videos with a length of 301 frames of the original data, as well as for 50 different generated videos with a length of 301 frames. We then computed the sample means and standard deviations of the correlation coefficients to compare our generated videos to the original ones. Pixels 1, 2, 3 and 4 used for comparison are given in Fig. 17.

The results in Fig. 4 indicate that, overall, the generated videos exhibit temporal dynamics consistent with the real videos, as the mean correlations almost always fall within one standard deviation of the real data. However, the model tends to exaggerate short-term correlations, particularly for points 1, 2 and 3. Whilst long-term correlations are captured effectively, there are notable deviations at late times at point 4, suggesting difficulty in replicating more non-periodic behaviours. An analysis of all four coordinate points and longer, 1000 frame videos is given in Section K.

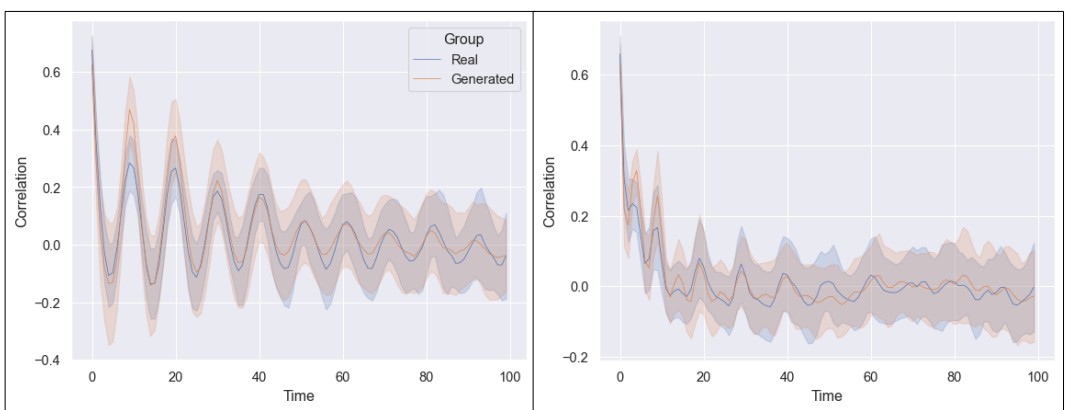

Figure 4: $\Delta t$ vs $\rho(\Delta t)$ with $\Delta t \in 1, \ldots, 100$ for pixel 1 (left), and pixel 3 (right). The shaded areas show a $1\sigma$ range. The mean and standard deviation of the correlation are calculated across 50 generated videos containing 301 frames, and for the real videos were across 100 real videos with 301 frames.

|  | Ours (full model) | | DeepONet | | FNO | |
|---|---|---|---|---|---|---|
|  | L1 | MSE | L1 | MSE | L1 | MSE |
| Heat (1) | **0.0370** | **0.0021** | 0.2754 | 0.1181 | 0.2753 | 0.1180 |
| Heat (200) | **0.0243** | **0.0009** | 0.3204 | 0.1603 | 0.2546 | 0.0996 |
| Heat (1000) | **0.0051** | **$4.27 \times 10^{-5}$** | Diverged | Diverged | Diverged | Diverged |
| Wave (1) | **0.0921** | **0.0133** | 0.2587 | 0.1090 | 0.2588 | 0.1090 |
| Wave (200) | 0.3181 | 0.1600 | 5.1761 | 2787.4028 | **0.2741** | **0.1186** |
| Wave (1000) | **0.4676** | **0.3420** | Diverged | Diverged | Diverged | Diverged |
| KSE (1) | **0.0095** | 0.0005 | 0.0259 | **0.0005** | 0.0190 | 0.0094 |

Table 1: Comparison of our model's performance to state-of-the-art neural operator methods across our datasets. Performance after 1, 200 and 1000 frames of continuous generation is given. Best results for each metric are highlighted in bold.

**Computational Cost.** The motivation to generate dynamical systems using GANs stems in part from their ability to significantly reduce computational costs at inference time compared to LES. Exact training times and hardware are listed for each model in Section I.

## 6 Conclusions and Limitations

**Conclusion.** We have introduced a system-theoretic operator learning framework for understanding when 'world models' can faithfully learn dynamics from tokenized observations. With observability, we explained when concatenated token histories suffice for reconstruction in both linear and nonlinear settings. Empirically, our sequence of models from analytic least-squares to full GANs validated this theory across four benchmark problems. In each case, world-model reconstructions achieved low prediction error and captured key temporal correlations. Compared to traditional LES, inference with our generative models is orders of magnitude faster. Compared to existing neural operator methods, our models perform much better for longer time horizons.

**Limitations.** Our analysis relies on a strong global observability assumption: the output map and underlying flow must admit a continuous, invertible reconstruction operator. In practice, certain coefficient choices violate observability under patch averaging. Moreover, accuracy of World Models can degrade over very long rollouts. We see this occur for the chaotic regimes, but this is not necessarily an issue since small variations in initial conditions lead to large variations later in time, a problem even the LES (and neural operators) suffer from. However, we still see a breakdown in the time correlation for longer roll outs, especially in the KSE case. Finally, since observability is a binary conditions, further research is required to characterise when exactly a dataset may satisfy the formal assumptions set out in Section 3, but in practice still be difficult to learn i.e. in some sense be 'close' to being non-observable.

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

## A EXTRANEOUS PROOFS

Here we give detail on some proofs of Section 3. We first formally define the empirical risk functions defined in Section 3.2:

$$\hat{\mathcal{L}}_{g,N,T}(\eta_g) = \frac{1}{N(\frac{T}{\Delta} - k - 1)} \sum_{j=1}^{N} \sum_{t \in \Gamma_T \setminus (\Gamma_{\Delta \bar{k}} \cup \{T\})} \|\eta_g(y_{\text{seq}}(t; x_0^{(j)})) - y(t + \Delta; x_0^{(j)})\|^2,$$

$$\hat{\mathcal{L}}_{G,N,T}(\eta_G) = \frac{1}{N(\frac{T}{\Delta} - k)} \sum_{j=1}^{N} \sum_{t \in \Gamma_T \setminus \Gamma_{\Delta k}} \|\eta_G(y_{\text{seq}}(t; x_0^j)) - x(t; x_0^{(j)})\|^2.$$

Here $T$ stands for the observation time of the trajectory, $\Delta$ for the time steps of the evolution and observation, and $N$ is the number of initial conditions $x_0^{(i)}$ sampled from some distribution $\mu$ on $\mathcal{X}$. They take their minimums when

$$\hat{\eta}_{g,N,T} \in \underset{\eta_g \in \mathcal{H}_{g,N,T}}{\arg\min} \hat{\mathcal{L}}_{g,N,T}(\eta_g), \quad \hat{\eta}_G \in \underset{\eta_G \in \mathcal{H}_{G,N,T}}{\arg\min} \hat{\mathcal{L}}_{G,N,T}(\eta_G), \tag{15}$$

Clearly,

$$\mathbb{E}_{x_0 \sim \mu}[\hat{\mathcal{L}}_{N,T,g}(\eta_g)] = \mathcal{L}_{T,g}(\eta_g) \text{ and } \mathbb{E}_{x_0 \sim \mu}[\hat{\mathcal{L}}_{T,G}(\eta_G)] = \mathcal{L}_{T,G}(\eta_G).$$

*Proof of Theorem 4.* We give the proof for the autoregressive map $g : \mathcal{Y}^k \to \mathcal{Y}$; the proof of PAC-learnability for $G : \mathcal{Y}^k \to \mathcal{X}$ is completely analogous.

We start with the usual error decomposition formula. Let $\eta_g \in \mathcal{H}_{g,N,T}$, then

$$0 \leq \mathcal{L}_g(\hat{\eta}_{g,N,T}) = \mathcal{L}_g(\eta_g) + \left[ \hat{\mathcal{L}}_{g,N,T}(\hat{\eta}_{g,N,T}) - \hat{\mathcal{L}}_{g,N,T}(\eta_g) \right]$$

$$+ \left[ \mathcal{L}_g(\hat{\eta}_{g,N,T}) - \hat{\mathcal{L}}_{g,N,T}(\hat{\eta}_{g,N,T}) \right] + \left[ \hat{\mathcal{L}}_{g,N,G}(\eta_g) - \mathcal{L}_g(\eta_g) \right]$$

$$\leq \mathcal{L}_g(\eta_g) + \left[ \hat{\mathcal{L}}_{g,N,T}(\hat{\eta}_{g,N,T}) - \hat{\mathcal{L}}_{g,N,T}(\eta_g) \right] + 2 \sup_{\tilde{\eta}_g \in \mathcal{H}_{g,N,T}} \left| \hat{\mathcal{L}}_{g,N,G}(\tilde{\eta}_g) - \mathcal{L}_g(\tilde{\eta}_g) \right|$$

$$\leq \mathcal{L}_g(\eta_g) + 2 \sup_{\tilde{\eta}_g \in \mathcal{H}_{g,N,T}} \left| \hat{\mathcal{L}}_{g,N,G}(\tilde{\eta}_g) - \mathcal{L}_g(\tilde{\eta}_g) \right|,$$

where the second term in the second last line is smaller or equal to zero by the ERM assumption (15). Taking the infimum of the right hand side over $\eta_g \in \mathcal{H}_{g,N,T}$, we obtain

$$0 \leq \mathcal{L}_g(\hat{\eta}_{g,N,T}) \leq \inf_{\eta_g \in \mathcal{H}_{g,N,T}} \mathcal{L}_g(\eta_g) + 2 \sup_{\eta_g \in \mathcal{H}_{N,T}} \left| \hat{\mathcal{L}}_{g,N,T}(\eta_g) - \mathcal{L}_g(\eta_g) \right|, \tag{16}$$

where the first term stands for the model error and the second for the generalization error.

Furthermore, the autoregressive map $g$ exists and is continuous by the representation given in the proof of Lemma 2.

Let $\delta, \varepsilon > 0$ arbitrary. Let $\mathcal{H}_\varepsilon$ be a space of fully connected neural networks of width $W(\varepsilon)$ and depth $L(\varepsilon)$ and with weights $\theta$ in weight space $\Theta_\varepsilon$ uniformly bounded by $M(\varepsilon)$ such that the first term on the right hand side of (16) is bounded by $\frac{\varepsilon}{2}$. Without loss of generality we can assume $\Theta_\varepsilon$ to closed and thus compact.

That this hypothesis space exists follows from the universal approximation property of neural networks, see e.g. Cybenko (1989); Yarotsky (2017); Kidger & Lyons (2020) which guarantee $\|\eta_g - g\|_\infty \leq \sqrt{\frac{\varepsilon}{2}}$ for some $\eta_g \in \mathcal{H}_\varepsilon$. From this and by (8a).

$$\mathcal{L}_{g,T}(\eta_g) \leq \|\eta_g - g\|_\infty^2 \quad \Rightarrow \quad \inf_{\eta_g \in \mathcal{H}_{g,N,T}} \mathcal{L}_g(\eta_g) \leq \frac{\varepsilon}{2}.$$

Not that all summands occurring in $\hat{\mathcal{L}}_{g,N,T}(\eta_g)$ are uniformly bounded for $\eta \in \mathcal{H}_\varepsilon$ and continuous in the neural networks' weights $\theta \in \Theta_\varepsilon$. By the uniform law of large numbers (Ferguson, 2017) it follows that the second term in (16) converges to zero $P$-almost surely

when $\mathcal{H}_{N,T}$ is replaced by $\mathcal{H}_\varepsilon$. As almost sure convergence implies convergence in probability, there exists a $N(\varepsilon, \delta)$ sufficiently large such that for $N \geq N(\varepsilon, \delta)$

$$P\left(\mathcal{L}_g(\hat{\eta}_{g,N,T}) > \varepsilon\right) \leq P\left(\sup_{\eta_g \in \mathcal{H}_{N,T}} \left|\hat{\mathcal{L}}_{g,N,T}(\eta_g) - \mathcal{L}_g(\eta_g)\right| > \frac{\varepsilon}{4}\right) \leq \delta.$$

Now the first inequality from Theorem 4 follows from passing over to the complementary events, provided we set $\mathcal{H}_{g,N(\varepsilon,\delta),T} = \mathcal{H}_\varepsilon$. □

Here we have given the proof of Theorem 4 in the large sample limit $N \to \infty$ for i.i.d. sampling of the initial conditions $x_0^{(j)} \sim \mu$. For ergodic dynamic systems, by application of Birkhoff (1931)'s ergodic theorem, this limit can be replaced by the large time limit $T \to \infty$ and only one observed trajectory $N = 1$, when $x_0 \sim \mu$ and $\mu$ is the invariant measure of the dynamic system, see Drygala et al. (2022) for a similar argument.

*Proof of Theorem 5.* We consider (10) and estimate

$$\mathcal{L}_S(\hat{\eta}_{g,N,T}, \hat{\eta}_{G,N,T})$$

$$= \mathbb{E}_{x_0 \sim \mu}\left[\|S_\Delta(x_0) - \hat{\eta}_{G,N,T}(y_{\mathrm{past}-}(x_0), \hat{\eta}_{g,N,T}(y_{\mathrm{past}}(x_0))\|^2\right]^{\frac{1}{2}}$$

$$= \mathbb{E}_{x_0 \sim \mu}\left[\|G(y_{\mathrm{past}-}(x_0), g(y_{\mathrm{past}}(x_0)) - \hat{\eta}_{G,N,T}(y_{\mathrm{past}-}(x_0), \hat{\eta}_{g,N,T}(y_{\mathrm{past}}(x_0))\|^2\right] \qquad (17)$$

$$\leq 3\mathbb{E}_{x_0 \sim \mu}\left[\|G(y_{\mathrm{past}-}(x_0), g(y_{\mathrm{past}}(x_0)) - \hat{\eta}_{G,N,T}(y_{\mathrm{past}-}(x_0), g(y_{\mathrm{past}}(x_0))\|^2\right]$$

$$+ 3\mathbb{E}_{x_0 \sim \mu}\left[\|\hat{\eta}_{G,N,T}(y_{\mathrm{past}-}(x_0), g(y_{\mathrm{past}}(x_0)) - \hat{\eta}_{G,N,T}(y_{\mathrm{past}-}(x_0), \hat{\eta}_{g,N,T}(y_{\mathrm{past}}(x_0))\|^2\right]$$

Here we used the triangular inequality for the $L^2$-norm and the fact that $\sqrt{c} \leq \sqrt{a} + \sqrt{b} \Rightarrow c \leq 3a + 3b$ for $a, b, c > 0$. In the next step we apply the substitution $x_0 \mapsto S_\Delta^{-k}(x_0) = x_0'$ we then obtain for the second term on the right hand side

$$\mathbb{E}_{x_0 \sim \mu}\left[\|\hat{\eta}_{G,N,T}(y_{\mathrm{past}-}(x_0), g(y_{\mathrm{past}}(x_0)) - \hat{\eta}_{G,N,T}(y_{\mathrm{past}-}(x_0), \eta_g(y_{\mathrm{past}}(x_0))\|^2\right]^{\frac{1}{2}}$$

$$\leq L_{\mathcal{H}_{G,N,T}} \mathbb{E}_{x_0 \sim \mu}\left[\|g(y_{\mathrm{past}}(x_0)) - \hat{\eta}_{g,N,T}(y_{\mathrm{past}}(x_0))\|^2\right]^{\frac{1}{2}}, \qquad (18)$$

where $L_{\mathcal{H}_{G,N,T}}$ is the maximal Lipshitz constant of neural networks $\eta_G$ in $\mathcal{H}_G$. Let us apply the change of variables $x_\mathsf{I} \to x_0' = S_\Delta^{-k}(x_0)$. Under this substitution we get

$$\mathbb{E}_{x_0 \sim \mu}\left[\|g(y_{\mathrm{past}}(x_0)) - \hat{\eta}_{g,N,T}(y_{\mathrm{past}}(x_0))\|^2\right]$$

$$= \mathbb{E}_{x_0' \sim (S_\Delta^{-k})_* \mu}\left[\|g(y_{\mathrm{seq}}(\Delta(k-1); x_0')) - \hat{\eta}_{g,N,T}(y_{\mathrm{seq}}(\Delta(k-1); x_0'))\|^2\right]$$

$$= \mathbb{E}_{x_0' \sim \mu}\left[\|g(y_{\mathrm{seq}}(\Delta(k-1); x_0')) - \hat{\eta}_{g,N,T}(y_{\mathrm{seq}}(\Delta(k-1); x_0'))\|^2 \frac{d(S_\Delta^{-k})_*}{d\mu}(x_0')\right] \qquad (19)$$

$$\leq \left\|\frac{d(S_\Delta^{-k})_* \mu}{d\mu}\right\|_\infty \mathbb{E}_{x_0' \sim \mu}\left[\|g(y_{\mathrm{seq}}(\Delta(k-1); x_0')) - \hat{\eta}_{g,N,T}(y_{\mathrm{seq}}(\Delta(k-1); x_0'))\|^2\right]$$

$$\leq \left\|\frac{d(S_\Delta^{-k})_* \mu}{d\mu}\right\|_\infty \left(\frac{T}{\Delta} - k\right) \mathcal{L}_{g,T}(\hat{\eta}_{g,N,T})$$

Here for a measurable set $A \subseteq \mathcal{X}$, $(S_\Delta^{-k})_* \mu(A) = \mu(S_\Delta^k(A))$ is the image probability measure of $\mu$ under $k$ steps in the backward dynamics and $\frac{d(S_\Delta^{-k})_* \mu}{d\mu}$ is the Radon-Nikodym derivative of $S_{(\Delta)_*}^{-k}\mu$ with respect to $\mu$.

By a similar calculation we obtain for the first term on the right hand side in (17)

$$\mathbb{E}_{x_0 \sim \mu}\left[\left\|G(y_{\text{past}-}(x_0), g(y_{\text{past}}(x_0))) - \hat{\eta}_{G,N,T}(y_{\text{past}-}(x_0), g(y_{\text{past}}(x_0)))\right\|^2\right]^{\frac{1}{2}}$$

$$\leq \left\|\frac{d(S_\Delta^{-k})_*\mu}{d\mu}\right\|_\infty \left(\frac{T}{\Delta} - k\right) \mathcal{L}_{G,T}(\hat{\eta}_{G,N,T}) \tag{20}$$

Let us assume that $\left\|\frac{d(S_\Delta^{-k})_*\mu}{d\mu}\right\|_\infty$ is finite and $\varepsilon, \delta > 0$ be given. We now choose hypothesis spaces and sample sizes according to the following recipe. Making use of the second assertion in Theorem 4, we see that the can choose $\mathcal{H}_{G,N,T}$ such that

$$P\left(\mathcal{L}_{G,T}(\hat{\eta}_{G,N,T}) > \frac{\varepsilon}{6\left\|\frac{d(S_\Delta^{-k})_*\mu}{d\mu}\right\|_\infty \left(\frac{T}{\Delta} - k\right)}\right) \leq \frac{\delta}{2}$$

$$\text{for } N \geq N_G\left(\frac{\varepsilon}{6\left\|\frac{d(S_\Delta^{-k})_*\mu}{d\mu}\right\|_\infty \left(\frac{T}{\Delta} - k\right)}, \frac{\delta}{2}\right).$$

By (20), we have that the first term on the right hand side of (17) is larger than $\frac{\varepsilon}{2}$ with probability at most $\frac{\delta}{2}$. We keep this hypothesis space for $G$ fixed and thus $L_{\mathcal{H}_{G,N,T}}$ is fixed as well. Using the first assertion in Theorem 4, we can further choose a hypothesis space and a natural number $N_g$ such that

$$P\left(\mathcal{L}_{g,T}(\hat{\eta}_{g,N,T}) > \frac{\varepsilon}{6\left\|\frac{d(S_\Delta^{-k})_*\mu}{d\mu}\right\|_\infty \left(\frac{T}{\Delta} - k\right)}\right) \leq \frac{\delta}{2} \tag{21}$$

$$\text{for } N \geq N_g\left(\frac{\varepsilon}{6L_{\mathcal{H}_{G,N,T}}\left\|\frac{d(S_\Delta^{-k})_*\mu}{d\mu}\right\|_\infty \left(\frac{T}{\Delta} - k\right)}, \frac{\delta}{2}\right). \tag{22}$$

Using this along with (18) and (19), we also see that the second term on the right hand side of (17) is larger $\frac{\varepsilon}{2}$ with probability at most $\frac{\delta}{2}$ and for $N \geq N_g$. Therefore, for $N \geq \max\{N_g, N_G\}$, we obtain the assertion of Theorem 5 from (17), the union bound for probabilities and equations (21) and (22).

To conclude our proof, we identify generic conditions under which $\left\|\frac{d(S_\Delta^{-k})_*\mu}{d\mu}\right\|_\infty$ is finite. To this aim let $d\mu(x) = f_\mu(x)\,dx$ on $\mathcal{X}$ where the density $f_\mu(x) > 0$ for all $x \in \mathcal{X}$ is continuous. As $\mathcal{X}$ is compact, $\kappa_{\min} = \min_{x \in \mathcal{X}} f_\mu(x) > 0$ and $\kappa_{\max} = \max_{x \in \mathcal{X}} f_\mu(x) < \infty$.

As already mentioned in Section 3.1, $S_{k\Delta} = (S_\Delta^{-k})^{-1}$ is Lipshitz by the Grönwall lemma. Thus, the Jacobian $DS_{k\Delta}(x)$ exists $dx$ almost everywhere on $\mathcal{X}$ and furthermore it is element wise essentially bounded by the global Lipshitz constant of $DS_{k\Delta}(x)$. Hence $\|\det(DS_{k\Delta})\|_\infty < \infty$.

From the change of variables formula of densities, we get

$$d(S_\Delta^{-k})_*\mu(x_0') = |\det(DS_{k\Delta}(x_0'))|\, f_\mu(S_{k\Delta}(x_0'))\,dx_0'.$$

Therefore,

$$0 \leq \frac{d(S_\Delta^{-k})_*\mu}{d\mu}(x_0') = \frac{|\det(DS_{k\Delta}(x_0'))|\, f_\mu(S_{k\Delta}(x_0'))}{f_\mu(x_0')} \leq \frac{\kappa_{\max}}{\kappa_{\min}}\|\det(DS_{k\Delta})\|_\infty < \infty. \qquad \square$$

*Proof of Theorem 6.* Let $\Gamma$ be the quadratic lattice with lattice constant $\delta > 0$, edge length $L$ and $\sqrt{k} = \frac{L}{\delta}$ lattice points on the edge. Eigenvectors of $A_h$ for $a = \text{const.}$ are those of the standard lattice Laplacian, which are well known. In fact, the trigonometric functions

$(\sin(\omega \cdot \xi))_{\xi \in \Gamma}$, $(\cos(\omega \cdot \xi))_{\xi \in \Gamma}$ diagonalise $A_h$ where $\omega = (\omega_1, \omega_2)$ and $\omega_j \in \{2\frac{\pi j \delta}{L} : j \in \{0, \ldots, \sqrt{k}-1\}\}$.

Let $\Lambda(\xi_t)$ be a square region in $\Gamma$ over which the tokenizer map $h$ averages. Let $q \in \mathbb{N}$, $1 < q < \sqrt{\kappa}$, be the length of the side of $\Lambda(\xi_t)$ measured in grid pixels. Suppose that $\omega_1$ or $\omega_2$ is from the set $\{\frac{2\pi j \delta}{L} : j = lq, l \in \mathbb{N}\}$. Then, $\cos(\omega \cdot \xi)$ and $\sin(\omega \cdot \xi)$ vanish when averaged over $\Lambda(\xi_t)$, as either in the first or the second direction these functions complete a full period. As this is valid for all centres of the token squares $\xi_t$, the projection $h$ sends these eigen functions to zero. From Theorem 3 (iii) it then follows that the dynamical system is not linearly observable.

The proof for the wave equation is similar, as $A_w$- eigen vectors of the form $(\sin(\omega \cdot \xi), \lambda(\omega) \sin(\omega \cdot \xi))$ are annihilated by the tokenizer by averaging over $\Lambda(\xi_t)$, if $\omega_1$ or $\omega_2$ are in the set $\{\frac{2\pi j \delta}{L} : j = lq, l \in \mathbb{N}\}$. Here $\lambda(\omega)$ the $A_h$-eigen value to $\sin(\omega \cdot \xi)$. $\qquad\square$

In the case that (1) defines an ergodic dynamical system with invariant measure $\mu$, we can replace the large sample limit in the number of trajectories $N(\varepsilon, \delta) \to \infty$ as $\varepsilon, \delta \downarrow 0$ with the large time limit $T(\varepsilon, \delta) \to \infty$ keeping $N = 1$ fixed, see Drygala et al. (2022) for this approach in a related situation based on Birkhoff's ergodic theorem.

## B  Continuous Time Control Theory

In continuous time control theory, Definition 1 (i) is changed by demanding $y(t; x_0) = y(t; x_0)$ for $t \in [0, T]$ instead of only for $t \in \Gamma_T$. Note that the dynamical system (1) is a special case of Hermann & Krener (1977) since the *distinguishing control* is assumed to be the trivial one. Let us emphasize that for nonlinear systems this generally diminishes otherwise existent observability properties, cf. (Casti, 1982, Section 5). On the other hand, considering uncontrolled systems has the advantage of characterizing observability equivalently in terms of the injectivity of the so-called *output* map

$$H \colon \mathcal{X} \to L^2(0, T; \mathbb{R}^m), \quad H(x_0) := h(x(\cdot; x_0)) = y(\cdot; x_0)$$

which assigns to a given initial state its corresponding output trajectory. As a consequence, observability of $(f, h)$ implies the existence of an inverse $H^{-1} \colon H(\mathcal{X}) \to \mathcal{X}$. Composing this function with the flow of (1) moreover allows us to define a *reconstruction map*

$$G \colon H(\mathcal{X}) \to L^2(0, \tilde{T}; \mathbb{R}^n), \quad G(y) := (S_{\tilde{T}} \circ H^{-1})(y) = x(\cdot; H^{-1}(y))$$

for any $\tilde{T} > 0$. Note that in the case of linear dynamics with a linear output map, the inverse $H^{-1}$ as well as the reconstruction map $G$ have the explicit representations

$$H^{-1}y = Q(\tilde{T})^{-1} \int_0^{\tilde{T}} e^{A^\top t} h^\top y(t) \, \mathrm{d}t, \quad Gy = e^{A \cdot} Q(\tilde{T})^{-1} \int_0^{\tilde{T}} e^{A^\top t} h^\top y(t) \, \mathrm{d}t$$

where the positive definiteness of the *observability Gramian* $Q(\tilde{T}) := \int_0^{\tilde{T}} e^{A^\top s} h^\top h e^{As} \, \mathrm{d}s$ is ensured by the observability of the system, see (Sontag, 1998, Section 6.3).

In Hermann & Krener (1977), other notions of observability have been discussed. Among those notions, the concept of *weak observability* relaxes classical observability in the sense of requiring invertibility of the output map only in a neighbourhood of a given state. One of the strengths of this notion is that it is particularly amenable to algebraic (rank) conditions which generalize the Kalman rank conditions for (7) available in the linear case. Let us briefly recall the discussions from Hermann & Krener (1977); Casti (1982) with slight adjustments to account for the simplified setup considered here and define the Lie derivative of a scalar (component) function $h_i \colon \mathcal{M} \to \mathbb{R}, i = 1, \ldots, m$ along the uncontrolled vector field $f \colon \mathcal{M} \to \mathcal{M}$ as

$$L_f h_i(x) = \nabla h_i(x)^\top f(x).$$

We then define $\mathcal{G}$ as the smallest vector space that is generated by the component functions $h_i$ and which is closed under iterated Lie differentiation along the vector field $f$. Given $x \in \mathcal{M}$ and denoting

$$\mathrm{d}\mathcal{G}(x) := \{z \in \mathbb{R}^n \mid z = \nabla \phi(x), \phi \in \mathcal{G}\},$$

we have the following well-known results, see (Hermann & Krener, 1977, Theorem 3.1, Theorem 3.12).

**Theorem 7.** *If the dynamical system* (1) *is analytic, then it is weakly observable at* $x_0 \in \mathcal{S}$ *under the observation map (tokenizer)* $h$ *if and only if* $\dim(\mathrm{d}\mathcal{G}(x_0)) = n$.

Unfortunately, the above rank conditions are not easily verifiable in all but the simplest examples. Moreover, the conditions only imply weak observability and therefore generally do not ensure invertibility of the output map, see (Hermann & Krener, 1977, Example 3.3). Even in the case of an invertible output map, explicit reconstruction formulas in terms of (nonlinear) observability Gramians do not exist in the general nonlinear case. On the other hand, it is known that observability is a *generic* property that is guaranteed to hold almost surely if arbitrarily smooth vector fields are considered (Aeyels, 1981). The latter reference further shows that observability can be achieved in a (practically relevant) setup where only finitely many output samples rather than a continuous output trajectory are available. In particular, it is shown that the number of samples has to be at least $2n + 1$ for an $n$-dimensional state space. Finally, we also point to Zeng (2018) which considers potential extensions of linear observability measures in terms of local generalisations of the observability Gramian and related cost functionals.

## C    COMPARISON TO OTHER METHODS

### C.1    IMPLEMENTATION DETAILS

The FNO used in the paper is developed by PhysicsNemo (Contributors, 2023) (formerly Modulus), a package developed by NVIDIA. The hyperparameters can be found in Table 2 below. The DeepONet consists of the same FNO architecture as the branch net, and a fully connected network as the trunk. Here, the FCN is built by us and connected to the branch net in the usual manner. Hyperparemeters are give in Table 3.

| Parameter | Value |
|---|---|
| Decoder layer size | 32 |
| Hidden channels | 64 |
| Fourier modes | 24 |
| No. of layers | 4 |
| Learning rate | 0.001 |
| Max epoch | 100 |

Table 2: The hyperparameters for the FNO model (Exact architecture given by NVIDIA's PhysicsNeMo package)

| Parameter | Value |
|---|---|
| Hidden channels | 150 |
| No. of layers | 3 |
| Learning rate | 0.001 |
| Max epoch | 100 |

Table 3: The hyperparameters for the trunk network of the DeepONet. The branch network was an FNO with the same architecture as in the FNO comparisons

|  | Ours (full model) | DeepONet | FNO |
|---|---|---|---|
| KSE (200) | **5.7109** | 6.1465 | 6.6150 |

Table 4: Results on the KSE dataset for the time correlation metric. The Euclidean distance between each point of the time correlation curves of the simulations and the model is measured and summed, up to 200 frames.

| Dataset | Condition | Frame | DeepONet | | FNO | | Ours | |
|---|---|---|---|---|---|---|---|---|
|  |  |  | L1 | MSE | L1 | MSE | L1 | MSE |
| Heat | eq | 1 | 0.01217 | 0.0002134 | **0.005122** | $\mathbf{5.544 \times 10^{-5}}$ | 0.1145 | 0.02102 |
| Heat | eq | 200 | 7.133 | 1397 | $7.175 \times 10^5$ | $4.264 \times 10^{12}$ | **0.2148** | **0.08548** |
| Heat | eq | 1000 | Diverged | Diverged | Diverged | Diverged | **0.669** | **0.7396** |
| Heat | point | 1 | **0.00401** | $\mathbf{2.559 \times 10^{-5}}$ | 0.005657 | $6.812 \times 10^{-5}$ | 0.04133 | 0.009935 |
| Heat | point | 200 | 10.74 | 2707 | **0.2952** | **0.1134** | 0.4876 | 0.4414 |
| Heat | point | 1000 | Diverged | Diverged | Diverged | Diverged | **1.525** | **3.84** |
| Heat | sin | 1 | 0.007225 | 0.0001037 | **0.004157** | $\mathbf{2.608 \times 10^{-5}}$ | 0.1241 | 0.02409 |
| Heat | sin | 200 | $4.219 \times 10^5$ | $8.944 \times 10^{11}$ | $1.6 \times 10^7$ | $1.242 \times 10^{15}$ | **0.03128** | **0.001517** |
| Heat | sin | 1000 | Diverged | Diverged | Diverged | Diverged | **0.01401** | **0.0003164** |
| Wave | eq | 1 | 0.02106 | 0.0004665 | **0.009501** | **0.0001064** | 0.04258 | 0.003717 |
| Wave | eq | 200 | **0.7729** | **0.6937** | $4.55 \times 10^7$ | $1.4 \times 10^{16}$ | 1.4 | 4.243 |
| Wave | eq | 1000 | Diverged | Diverged | Diverged | Diverged | **6.786** | **64.6** |
| Wave | point | 1 | 0.003903 | $2.37 \times 10^{-5}$ | **0.001271** | $\mathbf{3.376 \times 10^{-6}}$ | 0.0446 | 0.008845 |
| Wave | point | 200 | $5.508 \times 10^6$ | $2.211 \times 10^{15}$ | $7.155 \times 10^6$ | $3.807 \times 10^{14}$ | **1.026** | **17.74** |
| Wave | point | 1000 | Diverged | Diverged | Diverged | Diverged | **13.81** | **3353** |
| Wave | sin | 1 | 0.007625 | $8.853 \times 10^{-5}$ | **0.004253** | $\mathbf{3.937 \times 10^{-5}}$ | 0.04471 | 0.003131 |
| Wave | sin | 200 | $6.538 \times 10^5$ | $2.076 \times 10^{13}$ | $3.475 \times 10^6$ | $1.028 \times 10^{14}$ | **0.8098** | **1** |
| Wave | sin | 1000 | Diverged | Diverged | Diverged | Diverged | **3.57** | **19.5** |
| KSE | eq | 1 | **0.0111** | **0.000147** | 0.02004 | 0.0005884 | 0.2302 | 0.1113 |
| KSE | eq | 200 | 2.516 | 8.534 | **0.02438** | **0.0008629** | 0.3907 | 0.2384 |
| KSE | point | 1 | **0.01244** | **0.0001977** | 0.07488 | 0.01854 | 0.1859 | 0.1057 |
| KSE | point | 200 | 2.584 | 10.24 | **2.197** | **8.133** | 0.3008 | 0.1318 |
| KSE | sin | 1 | **0.0111** | **0.000147** | 0.02006 | 0.0005886 | 0.1696 | 0.1217 |
| KSE | sin | 200 | 2.514 | 8.527 | **0.02792** | **0.001137** | 0.2992 | 0.132 |

Table 5: Performance comparison between DeepONet, FNO, and our model across datasets, initial conditions, and generation horizon (frame). Best values in each row are highlighted in bold.

## C.2 Other Comparison Metrics

Table 5 compares the performance of the models on OOD initial conditions. We tested on three OOD initial conditions:

1. The field $z = (x(1-x)y(y-1))^2$, $\quad 0 < x, y < 1$, with $z$ normalised to $[-1, 1]$, labelled as 'eq' in the table. This gives a smooth, non-trivial initial condition with zero values at the boundaries of the domain;

2. A single central point source 'point', where the centre of the grid is set to 1 and everywhere else left as 0. This tests a highly discontinuous initial condition;

3. A sin wave 'sin', given by $z = \sin(2\pi x)\sin(2\pi y)$, $\quad 0 < x, y < 1$

All three conditions are evaluated on a meshgrid, discretised to the resolution required by the dataset. The initial conditions are evolved using the same numerical techniques given in Section E.

Meanwhile, Table 4 shows a metric measuring the distance between the correograms (x-axis 1-200 frames) of the model and ground truth data, for both the DeepONet and FNO.

Our model is the most consistent, but is occasionally beaten on the 1-frame ahead metrics. However, as the models roll out to further time horizons, our model is generally the best. As with the in-distribution metrics, none of the alternative models survive to 1000 frames.

### C.3 Ablation Study for the FNO

| Model | Correlation at $t = 50$ | Correlation at $t = 200$ | Correlation at $t = 400$ |
|---|---|---|---|
| Ours | 2.3749 | **5.6150** | **11.9308** |
| $(2, 8, 12, 16)$ | 1.7484 | 6.3689 | 14.6852 |
| $(2, 8, 24, 16)$ | 1.6072 | 6.2392 | 14.8404 |
| $(2, 8, 12, 32)$ | 1.5180 | 6.3100 | 14.9246 |
| $(2, 8, 24, 32)$ | 1.6583 | 6.4774 | 14.8132 |
| $(3, 16, 24, 32)$ | 1.5537 | 5.6931 | 14.5094 |
| $(3, 16, 24, 16)$ | 1.5214 | 6.2221 | 14.8424 |
| $(2, 16, 24, 16)$ | 1.5779 | 5.6759 | 14.7714 |
| $(2, 16, 12, 32)$ | 1.8459 | 6.1103 | 14.4958 |
| $(2, 16, 24, 32)$ | 1.6168 | 6.3857 | 15.3323 |
| $(3, 16, 12, 32)$ | **1.5153** | 6.1225 | 14.7511 |

Table 6: Results for the ablation study on FNO architecture of the best 10 models, across three different time points, as measured by the difference between the original data correogram and the correogram of the model.

| Setting | Values |
|---|---|
| No. of layers | $[1, 2, 3, 4, 5]$ |
| Decoder layer size | $[8, 16, 32, 64]$ (if 1 layer use upper half; if 5 layers use lower half) |
| Fourier modes | $[12, 24, 48]$ |
| Hidden channels | $[16, 32, 64]$ |

Table 7: Explored hypereparameter ranges for the FNO ablation study

Table 6 compares the Euclidean distance between the correogram of the model and the data for the KS equation, as done in Table 4. In total, we trained 144 additional models, of which we picked the best 10, discriminated by performance on the validation set. Models above a threshold validation loss at epoch 25 were stopped early.

Models are written in the form (no. of layers, decoder layer size, Fourier modes, hidden channels). The results of the ablation study show that a somewhat smaller model is the most effective, with the best at $t = 200$ being $(3, 16, 24, 32)$ and at $t = 400$ being $(2, 16, 12, 32)$, other than our model. This table reinforces our claim that our approach is much more effective for longer time horizons. The exact searched parameter space is given in Table 7.

## D Autoregressive Training with GANs

Adversarial training methods are often superior to regression by the fidelity of fine grained details. The idea behind adversarial training is the variational representation of norms or other measures of divergence. In the simplest case, consider two vectors $a, b \in \mathbb{R}^m, a \neq b$, then by the Cauchy-Schwartz inequality $\|a - b\| = \sup_{\mathbf{v} \in \mathbb{R}^m, \|v\|=1} v \cdot (a - b)$.

Let us consider the slightly modified risk functions for the autoregressive update step

$$\mathcal{L}_{g,T}^v(\eta_g) = \mathbb{E}_{x_0 \sim \mu, t \sim U(\Gamma_T \setminus (\Gamma_{\Delta(k-1)} \cup \{T\}))} \left[ \| \eta_g(y_{\text{seq}}(t, x_0)) - y(t + \Delta; x_0) \| \right]$$

$$= \mathbb{E}_{x_0 \sim \mu, t \sim U(\Gamma_T \setminus (\Gamma_{\Delta(k-1)} \cup \{T\}))} \left[ \sup_{v \in \mathbb{R}^m, \|v\|=1} v \cdot (\eta_g(y_{\text{seq}}(t; x_0)) - y(t + \Delta; x_0)) \right]$$

$$= \sup_{v_g : \mathcal{Y}^{\times k} \to \mathbb{R}^m} \mathbb{E} \left[ v_g(y_{\text{seq}}(t, x_0)) \cdot (\eta_g(y_{\text{seq}}(t : x_0)) - y(t + \Delta; x_0)) \right].$$

where the last step is possible as $v_g = \text{norm}(\eta_g(y_{\text{seq}}) - g(y_{\text{seq}}))$ is an explicit point wise maximizer, where $\text{norm}(v) = \frac{v}{\|v\|}, v \in \mathbb{R}^m \setminus \{0\}$. Note that here again we used existence of $g$ and hence observability of the dynamic system. The idea of adversarial training is to learn the function $v_g$ with a neural network from a sufficiently large hypothesis space $\mathcal{H}_{g,v,T}^v$ of neural networks with norm activation in the last layer. Consider the min-max optimisation problem

$$\eta_g \in \argmin_{\eta_g \in \mathcal{H}_{g,v,T}} \max_{\eta_g^v \in \mathcal{H}_{g,v,T}^v} \mathbb{E}_{x_0 \sim \mu, t \sim U(\Gamma_T \setminus (\Gamma_{\Delta(k-1)} \cup \{T\}))} \left[ \eta_g^v(y_{\text{seq}}(t; x_0)) \cdot \eta_g(y_{\text{seq}}(t; x_0)) \right]$$

$$- \mathbb{E}_{x_0 \sim \mu, t \sim U(\Gamma_T \setminus (\Gamma_{\Delta(k-1)} \cup \{T\}))} \left[ \eta_g^v(y_{\text{seq}}(t; x_0)) \cdot y(t + \Delta; x_0) \right].$$

As $y_{\text{seq}}(t; x_0)$ can't be solved for all $x_0 \in \mathcal{X}$, we sample $x_0^{(j)} \sim \mu$, $j = 1, \ldots, N$, and average over $t$ and $j$ to obtain the empirical risk function

$$\hat{\mathcal{L}}_{g,N,T}^v(\eta_g, \eta_g^v) = \frac{1}{N \left( \frac{T}{\Delta} - k \right)} \sum_{j=1}^{N} \sum_{t \in \Gamma_T \setminus (\Gamma_{k\Delta} \cup \{T\}))} \left[ \eta_g^v(y_{\text{seq}}(t; x_0^{(j)})) \cdot \eta_g(y_{\text{seq}}(t; x_0^{(j)})) \right.$$

$$\left. - \eta_g^v(y_{\text{seq}}(t; x_0)) \cdot y(t + \Delta; x_0) \right]$$

Given $\hat{\mathcal{L}}_g^v$, the training of the 'critic' network $\hat{\eta}_g^v$ and the generator $\hat{\eta}_g$ is done in the usual adversarial fashion

$$\hat{\eta}_{g,N,T}^v(\eta_g) \in \argmax_{\eta_g^v \in \mathcal{H}_{g,N,T}^v} \hat{\mathcal{L}}_{g,N,T}^v(\eta_g, \eta_g^v)$$

$$\hat{\eta}_{g,N,T} \in \argmin_{\eta_g \in \mathcal{H}_{g,N,T}} \hat{\mathcal{L}}_{g,N,T}^v(\eta_g, \hat{\eta}_g^v(\eta_g)) = \argmin_{\eta_g \in \mathcal{H}_{g,N,T}} \max_{\eta \in \mathcal{H}_{g,N,T}^v} \hat{\mathcal{L}}_{g,N,T}^v(\eta_g, \eta_g^v)$$

For $\eta_G$ we can set up an analogous adversarial training scheme. Using the decomposition for adversarial learning (Asatryan et al., 2020; Biau et al., 2021; Drygala et al., 2025) an adversarial treatment of PAC-learning is feasible as well. This however goes beyond the scope of this article. We also note that in the LongVideoGAN (Brooks et al., 2022) experiments, adversarial training is performed with respect to a 'discriminator' instead of a 'censor' network, see Arjovsky et al. (2017) for the difference.

# E   DISCRETIZATION OF THE PDE

In this section we aim to discretise the PDEs (Eqs. 11 and 13) and give an account of our numerical solvers.

## E.1   LINEAR EQUATIONS

We discretize the spatial domain using a uniform Cartesian grid with periodic boundaries and spacing $\Delta x$ in both the $x$- and $y$-directions. Fix the total number of grid steps in each direction as $n$. Let $u_{i,j}^\tau = (u_{i,j})_k^\tau = u_k^\tau$ with $k = ni + j$ denote the numerical approximation of $u(x_i, y_j, t^\tau)$, where $x_i = i\Delta x$, $y_j = j\Delta x$, and $t^\tau = \tau \Delta t$.

The left and right discrete derivatives are (in the $x$-direction) (Strikwerda, 2004)

$$D_x^+ u_{i,j} := \frac{u_{i+1,j} - u_{i,j}}{\Delta x}, \quad D_x^- u_{i,j} := \frac{u_{i,j} - u_{i-1,j}}{\Delta x}$$

which are linear maps and therefore have matrix representations. Then to form the modified Laplacian we write

$$
\begin{aligned}
(\nabla \cdot (a\nabla)\, u)_i &= \frac{\partial}{\partial x}\left(a\frac{\partial}{\partial x}\right)u + \frac{\partial}{\partial y}\left(a\frac{\partial}{\partial y}\right)u \\
&\approx \sum_{j=1}^{n^2}\left[\sum_{k=1}^{n^2}\left(D_x^-\right)_{ik}a_k\left(D_x^+\right)_{kj} + \sum_{k=1}^{n^2}\left(D_y^-\right)_{ik}a_k\left(D_y^+\right)_{kj}\right]u_j \\
&:= \sum_{k=1}^{n^2}\left(A_h\right)_{ik}u_k
\end{aligned}
\tag{23}
$$

For an appropriately generated positive definite vector $a$; see Section G. In the case $a$ is a vector of ones, this expression reduces to the standard five-point stencil. We write $u_{i,j}$ which lives in the torus defined on $[-1,1]^2$ as a vector $u_k$, with dimension $n^2$, and as such a mapping (with appropriately wrapping indices) $(i,j)\mapsto(ni+j)$ is required to perform the matrix multiplications defined in Eq. 23.

Applying the explicit (forward) Euler method for the time derivative:

$$
\frac{u_{i,j}^{\tau+1} - u_{i,j}^\tau}{\Delta t} = A_h u_{i,j}^\tau,
$$

we obtain the update formula:

$$
u_{i,j}^{\tau+1} = \Delta t\left(A_h u_{i,j}^\tau\right) + u_{i,j}^\tau
$$

Periodic boundary conditions are imposed by wrapping indices modulo the grid size. Numerical parameters $(\Delta x, \Delta t)$ are listed in Table 8.

The discretisation for the wave equation is similar, but instead requires the first order formulation 12, which gives the update rule for the full state space $u_{i,j}' = (u,v)_{i,j}$ as

$$
u_{i,j}'^{\tau+1} = \Delta t\left(A_w u_{i,j}'^\tau\right) + u_{i,j}'^\tau, \quad A_w = \left(\begin{array}{c|c} 0 & \mathbb{1} \\ \hline A_h & 0 \end{array}\right)
$$

Again, relevant numerical parameters can be found in Table 8.

### E.2 Nonlinear Equations

The KS dataset was generated by numerically integrating the governing partial differential equation 13 using a fourth-order Exponential Time Differencing Runge-Kutta method (ETDRK4), as described in Kassam & Trefethen (2005). This method is particularly well-suited for stiff PDEs such as the KS equation due to its ability to handle the linear stiff components analytically while treating the nonlinear terms explicitly. The spatial discretization was performed using Fourier spectral methods under periodic boundary conditions. The scheme uses a precomputed set of coefficients derived from contour integrals in the complex plane, which helps resolves some numerical instability issues.

The large-eddy simulation (LES) dataset was generated on a computational fluid dynamics (CFD) mesh and subsequently interpolated to a uniform compute grid. The LES resolves the large-scale turbulent structures explicitly while modelling the effect of unresolved subgrid-scale motions through a suitable turbulence closure. The interpolation step ensures that the data is represented on a regular grid compatible with spectral or finite-difference analysis, and can be represented in pixel-space for training with machine-learning methods.

## F  LOCAL OBSERVABILITY OF THE KS EQUATION

Even local observability according to Theorem 7 is hard to assess for the given equation, as for the minimal compression factor applied $k = 16$ this would require the computation of 16-th order lie derivatives. However, we give some numerical evidence for the simpler, one dimensional KSE with compression factor of 5.

For Theorem 7, we will verify a sufficient condition for local weak observability by selecting a subset of finitely many iterated Lie derivatives $L_f^l h_i$ for $k = l, \ldots, k-1$, approximating each derivative via finite differences of the sampled outputs, assembling the resulting gradients into a numerical observability matrix $\mathcal{O}(x_t)$, and confirming that it has full rank, ensuring $\dim\big(\mathrm{d}\mathcal{G}(x_t)\big) = n$ and hence local weak observability.

The KSE (13) in one spatial dimension $\frac{\partial u}{\partial t} + u \frac{\partial u}{\partial x} + \frac{\partial^2 u}{\partial x^2} + \frac{\partial^4 u}{\partial x^4} = 0$ with periodic boundary conditions is solved via the ETDRK2-Method (Cox & Matthews, 2002). We define a tokenized state $y(t) \in \mathbb{R}^m$ by averaging $u(x,t)$ over sliding spatial windows with stride $p = 5$. After an initial buffer of snapshots sufficient to approximate Lie derivatives up to order $p = 5$ via forward finite differences, the observability matrix $\mathcal{O}(t_k)$ is constructed by stacking these Lie-derivative approximations.

This yields a local linear approximation of the observability matrix $\mathcal{O}(x_t)$ at $x_t$ whose sign-preserving $\log \det(\mathcal{O}(x_t))$ remains finite, indicating full rank of the observability matrix $\mathcal{O}(x_t)$ over the time interval and hence local weak observability of the dynamical system after a burn-in period which can be interpreted as the time of transition from an initially almost constant state to the onset of the chaotic dynamics.

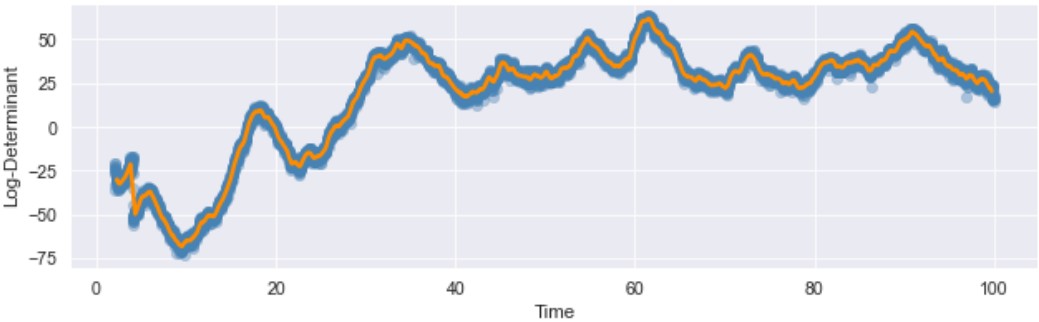

Figure 5: Time trace of $\log(\det \mathcal{O}(x_t))$ and its centered 50-step rolling average. Here $L = 80$, $u(x,0) = \sin(7\pi x/L)$; spectral discretization with $N = 200$ modes and ETDRK2 time stepping ($\Delta t = 0.01$, $T = 10000$ steps); coarse state $y \in \mathbb{R}^{N/p}$ via averaging over sliding spatial windows of size $p = 5$ ($N/p = 40$); first $p = 5$ time-derivatives by forward differences. The bounded $\log \det$ indicates $\mathrm{rank}\,\mathcal{O}(x_t) = N$.

## G  FURTHER DATASET DETAILS

Here we list the additional details important to the production of the datasets, but too superfluous for the main text.

### G.1  LINEAR EQUATIONS

The heat and wave equation solutions are constructed by forward Euler, an explicit method. This scheme, while simple, is known to converge for appropriately small time steps for these equations (LeVeque, 2007). Since the Laplacian $\nabla^2$ is not observable under the tokenization operator (Theorem 6) we generate a modified Laplacian $\nabla(a\nabla)$ where $a$ is a stochastically generated positive definite diagonal matrix. Examples of the low-res (super-res) heat (wave) dataset can be found in the left hand column of Fig. 8 (Fig. 9), in the Appendix. The

| Name | Heat (low-res) | Heat (super-res) | Wave (low-res) | Wave (super-res) |
|---|---|---|---|---|
| Resolution | $32 \times 32$ | $128 \times 128$ | $32 \times 32$ | $128 \times 128$ |
| Compression factor | 16 | 16 | 16 | 16 |
| History size | 16 | 7 | 16 | 7 |
| No. of init conditions | 100 | 75 | 100 | 75 |
| Frames (per init. cond.) | 2000 | 1000 | 2000 | 1000 |
| $\Delta t$ | 0.4 | 0.4 | 0.05 | 0.05 |

Table 8: Summary of the parameter settings for the heat and wave datasets. All datasets in this table used a test-train split ratio of 0.9, and a grid step ($\Delta x$) of 1.

| Name | KSE (low-res) | KSE (super-res) |
|---|---|---|
| Resolution | $64 \times 64$ | $256 \times 256$ |
| Compression factor | 16 | 16 |
| History size | 16 | 7 |
| Frames (per init. cond.) | 2000 | 2000 |
| No. of init. cond. (train) | 28 | 28 |
| No. of init. cond. (test) | 5 | 5 |
| Frame skip | 10 | 10 |
| Burn-in frames | 500 | 500 |
| $\Delta t$ | 0.01 | 0.01 |

Table 9: Parameter settings of the KSE datasets.

compression factor was selected to roughly match that of the LongVideoGAN model. Table 8 gives further details on the production of these datasets.

These datasets are generated on the flat torus $\mathbb{T}^2 = [-1, 1]^2$ by employing index wrapping i.e. circular boundary conditions. For all linear datasets, the data is linearly normalised to the range $[-1, 1]$. The seeds used to generate $a$ for the operator $\nabla(a\nabla)$ can be found in the supplementary material, published with the paper. As noted in B, observability is a generic condition, and so we have observability for our modified operator almost surely.

### G.2  KS Equation

A large amount of data was created by adopting the ETDRK4 (an exponential 4th order Runge-Kutta method) (Kassam & Trefethen, 2005), which we modified to produce solutions to the 2D equation. Notably, the 0th mode in Fourier space was zeroed out to prevent the growth of the mean $\bar{u}$ (see Eq. 2.2 in Kalogirou et al. (2015)). The precise numerical parameters used can be found in Table 9, and still frames from the super-res dataset can be found in the left column in Fig. 7.

For each initial condition, 25000 time steps are simulated, keeping all but 1 in 10. The first 500 frames are then discarded as burn-in, leaving each initial condition with 2000 frames for use in the dataset. The initial conditions were selected so that the resulting solution displays the desired chaotic behaviour. The low-res dataset is obtained by applying a low-resolution mask to the super-res set.

### G.3  Initial Conditions

Initial conditions $u_0$ are generated on the flat torus $\mathbb{T}^2 = [-1, 1]^2$ via samples from a mean-zero Gaussian random field with a Matérn kernel adapted to periodic boundary conditions.

| Parameter | Value |
|---|---|
| Noise type | Normal |
| Noise strength $\sigma$ | 10 |
| Inverse correlation length scale $m$ in lattice step units | 0.1 |
| Smoothness of Matern function $\nu$ | 1 |

Table 10: Settings for generating Gaussian random fields for the initial conditions

The covariance function is defined

$$\mathbb{E}[u_0(x)u_0(x')] = \frac{1}{4} \sum_{\kappa \in \pi\mathbb{Z}} \frac{e^{i\kappa \cdot (x-x')}}{(|\kappa|^2 + m^2)^\nu},$$

where $x, x' \in \mathbb{T}^2$, $m > 0$ is the inverse correlation length scale and $\nu > 0$ is the smoothness parameter. Table 10 summarizes the values of $(\sigma, m, \nu)$ used in the datasets. The strength of the noise depends linearly on $\sigma$. This is a standard model of non trivial spatial random structures that is widely used in e.g. geo-statistics or the theory of PDEs with random coefficients. The resulting Gaussian random field is then transformed with the exponential function to give positive values for the conductivity. This is implemented using the fast Fourier transform on the discrete torus: First FTT the noise, then multiply with the FFT of the square root of the Matern covariance operator, then apply the inverse FFT, and then exponentiate.

### G.4 Flow Around a Cylinder

For the experiments involving flow around a cylinder at Reynolds number 3900, we used a publicly available dataset Winhart & di Mare (2024) provided by the authors of Drygala et al. (2022). The dataset consists of 100,000 greyscale images with a resolution of $1000 \times 600$ pixels, capturing the characteristic Kármán vortex street - a coherent structure of alternating vortices aligned with the cylinder's axis. The data was generated using Large Eddy Simulation (LES), and the resulting unsteady velocity fields were processed via a projection mapping technique that preserves ergodicity (Peters, 2019) in a reduced state space. For details on the LES numerical setup, we refer to Drygala et al. (2022). The details on the setup and configuration of neural network training can be found in Section H.

## H Setup and Configuration of Neural Network Training

### H.1 Linear Equations

Here, a single autoregressive linear layer is sufficient, with input and output sizes matching the relevant dataset for each model. This has the added benefit of entirely preventing over-fitting, since the number of degrees of freedom in the model matches exactly that of the reconstruction map, when the history size is set to the compression factor, assuming there is enough data to properly constrain the model. We used the Adam optimiser (Kingma & Ba, 2017), and an ablation study was conducted to verify the optimality of the hyperparameters, albeit with a coarse grid since the purpose of these examples is to empirically substantiate the learning of the time evolution operator and confirm the theory, and each run represents significant computational expense (I). A learning rate of $1 \times 10^{-5}$ was used, and the number of steps given in Section I. We employed Mean Squared Error (MSE) as the objective function during training. Such a low learning rate and high number of steps is required due to the ill-conditioned nature of these problems.

## H.2 KS EQUATION

| Parameter | Value |
|---|---|
| Features per block (Low-res) | $[64, 128, 256, 512]$ |
| Features per block (Super-res) | $[64, 128, 256]$ |
| Learning rate | 0.0005 |
| Batch size | 768 |

Table 11: Hyperparameter settings for KSE models

The KSE dataset was trained with a UNet in a GAN-like fashion. The generator comprises four double 3D convolution blocks for the low-res case, which is the encoder, a bottleneck (itself a double convolution block), and then five 3D double transpose convolution blocks serve as the encoder. Between each block in the encoder and the respective block in the decoder there exists a skip connection. The super-res case is the same, but with one less block in the decoder and the encoder. A rough visual representation of the architecture is given in Fig. 6. Both models were trained adversarially, but with a single tensor in the shape of the model output as the discriminator network; equivalently, a neural network with a single linear layer and without a bias vector. Further hyperparameters are listed in Table 11.

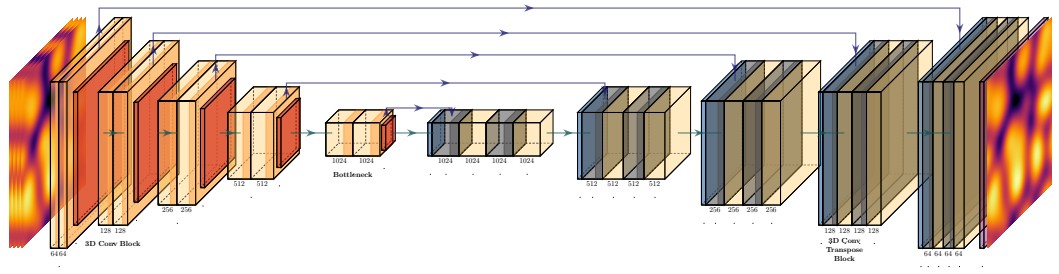

Figure 6: Architecture of the UNet in the low-res case, for the KS model. The very last layer is a final convolution to send the output from history size × resolution → resolution. All layers use a ReLu activation, and batch normalisation.

## H.3 FLOW AROUND A CYLINDER

NVIDIA's LongVideoGAN (Brooks et al., 2022) is designed to generate realistic video sequences with temporal consistency. Its architecture comprises two key components: a low-resolution GAN that captures time dynamics and a high-resolution GAN that enhances vortex detail by upscaling turbulent flows synthesised by the low-resolution GAN. Together, they ensure both the plausibility of individual frames and coherent motion throughout the video. The low-resolution GAN generates video sequences with resolution $64 \times 36$ pixels. It starts by sampling an 8-dimensional latent vector for each frame from a standard Gaussian distribution. To enforce temporal consistency, these vectors are smoothed using Kaiser low-pass filters (Kaiser, 1974), which reduce high-frequency noise while preserving long-term patterns. Each value in the filtered sequence is influenced by the entire sequence, including future frames. After smoothing, the vectors are processed through fully connected and 3D convolutional layers to produce low-resolution frames that capture the essential dynamics and structure of the video. The high-resolution GAN upsamples the low-resolution video to resolutions such as $1024 \times 576$ pixels, using the latter as conditional input. To generate the $n$-th high-resolution frame, it conditions on frames $n - 4$ to $n + 4$ from the low-resolution sequence, preserving temporal consistency. Aside from this conditioning, its architecture closely follows that of StyleGAN3 (Karras et al., 2021), incorporating convolutional and upsampling layers. During training, it is conditioned on real low-resolution videos aligned with the target high-resolution outputs. DiffAug (Zhao et al., 2020) is applied to improve robustness through augmentations such as translation and cutout, while R1 regularisation

(Mescheder, 2018) stabilises training by penalising the discriminator only on real samples. Once trained, the low-resolution GAN generates a video, which is then refined and upscaled by the high-resolution GAN to produce the final output.

**Temporal Correlation Metric**  To assess how well VideoGAN and the KSE model capture turbulent flow dynamics, we use a temporal correlation metric. For a video of length $T$, we compute the Pearson correlation $\rho(\Delta t)$ between the values of a specific pixel at time $t$ and $t + \Delta t$ for all $t \in 1, \ldots, T - \Delta t$. This measures how predictive a pixel's value is over time. High $\rho(\Delta t)$ indicates strong temporal consistency. Additional evaluation metrics specific to VideoGAN are provided in Appendix K.

**Training Configuration**  NVIDIA's LongVideoGAN requires a $16 : 9$ image format. To reshape the data to $1024 \times 576$ resolution, we added 24 white columns to the left side of the image and removed 12 rows from both the top and bottom. These images were then downscaled to $64 \times 36$ using Python's `Pillow` library for low-resolution input. Grayscale images were converted to RGB by replicating the single channel three times. The resulting $19,990$ images were split into 11 directories, each forming a non-overlapping video of approximately $1,817$ frames for training. Both GANs were trained on 2 NVIDIA A100 GPUs with a batch size of 8 and gradient accumulation of 1. All other hyperparameters followed those from the original LongVideoGAN paper and code.

# I  COMPUTATIONAL COST

## I.1  LINEAR EQUATIONS

All models were trained on a single NVIDIA A100 GPU. The heat equation models ran for approximately 54,000 epochs (low resolution, 10 hours 10 minutes) and 42,000 epochs (high resolution, 21 hours 58 minutes). Generating the low-res heat dataset took 1 hour 12 minutes, whereas the super-res dataset took only 17 minutes 30 seconds. The wave equation models ran for approximately 62,000 epochs (low resolution, 11 hours 45 minutes) and 35,000 epochs (high resolution, 18 hours 16 minutes), and the low-res wave dataset took 3 hours 35 minutes to produce, whilst the super-res dataset took 59 minutes. We note that we did not use a performance tuned code to create these datasets, and in particular, sparse multiplications could speed up these times significantly. However, of more relevance to a real world application are the inference times. Since the models for both low-res datasets are linear layers of the same size, they report similar inference times of $0.0002s$ per frame (averaged across 10 batches of 512 frames), vs the $0.0216s$ per frame for the heat and $0.0645s$ for the wave. That represents a speed up of about a factor of 100 in the heat case and even more for the wave case. The super-res models take slightly longer at $0.0007s$, averaged over 10 batches of 256 frames, vs a generation time of $0.014s$ in the heat case and $0.0472s$ in the wave case. This is again an orders of magnitude speed up.

## I.2  NON-LINEAR EQUATION

The KSE models were also trained on a single NVIDIA A100 GPU. The low-resolution model ran for approximately 40,000 steps and was stopped after 17 hours 49 minutes, while the high-resolution model ran for 18,000 steps and took 22 hours 46 minutes. In a similar vein as the linear datasets, we see an orders of magnitude speed up at inference time. The dataset used for both the low- and super- res case, which took a total time of 3 hours, 58 minutes. The low-resolution model averages an inference time of $0.012s$ per frame, across 10 256-frame batches. We compare this to a time of $0.216s$ per frame to produce the dataset. Again, we see a speed up of over 18x, and this time, the data generation method is highly optimised.

## I.3  LARGE EDDY SIMULATION

The LES was run on Intel Xeon "Skylake" Gold 6132 CPUs running at 2.6 GHz and equipped with 96 GB of RAM. The simulation used 20 nodes, each with 28 cores, and ran for approximately 20 days, for a total of $1,440$ core weeks (Drygala et al., 2022).

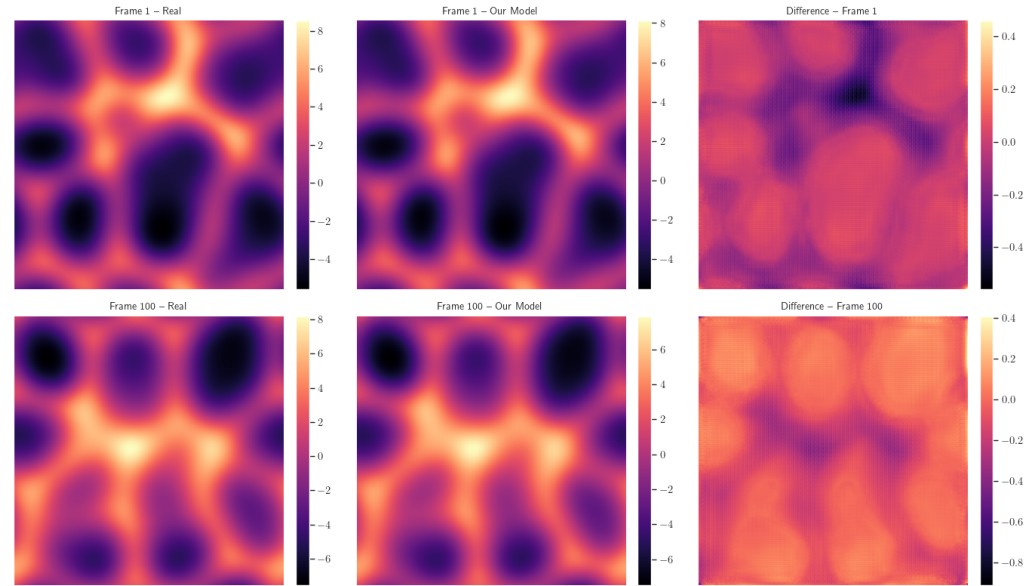

Figure 7: Examples from the KSE dataset, complete with the model predictions. This data is unseen by the model at training time. Note the relatively quick divergence from the numerical solver when generating forward in time, due to the chaotic nature of the equation.

### I.4 LONGVIDEOGAN

Our training of the Video GAN using 2 A100 GPUs took about 7 days, and the generation of a 1000 frame video takes about 32 seconds using one NVIDIA R6000. (On the Anonymous University HPC cluster it then takes another 60 seconds to save a video of length 1000.) The generation process is GPU RAM intensive and will require more than one GPU to generate videos with more than $10,000$ frames and/or higher resolution. We note that GANs are particularly fast for this kind of dataset, see Drygala et al. (2024) for more details. At 20 days for 100,000 frames, the CFD simulation produces 1 frame approximately every $17s$. Our model produces 1 frame every $0.032s$, which is a speed up factor of over 500x. We emphasise that this speed up is precisely the point of these experiments, and CFD simulations remain prohibitively expensive.

## J DATASET SNAPSHOTS

In this section we provide some snapshots of the datasets, together with respective result of the model, and finally the difference between the two. All examples are taken from the test sections of the dataset. Fig. 7 starts us with some examples from the super-res KS dataset, with the model producing results via the full pipeline, that is, as with Fig. 3. Fig. 8 also shows results for the super-res full model dataset, but for the heat equation, and Fig. 9 are examples from the low-res wave dataset. Finally, we show a real and generated example from the dataset of the flow around the cylinder in Fig. 10.

## K FURTHER EVALUATIONS

### K.1 UNMODIFIED WAVE EQUATION

We also trained on the non-observable (unmodified) wave equation and standard tokenizer pair. All other parameters remained the same. As Fig. 11 demonstrates, we again obtain significantly reduced performance when compared to the observable (modified equation) case, as seen in Fig. 2.

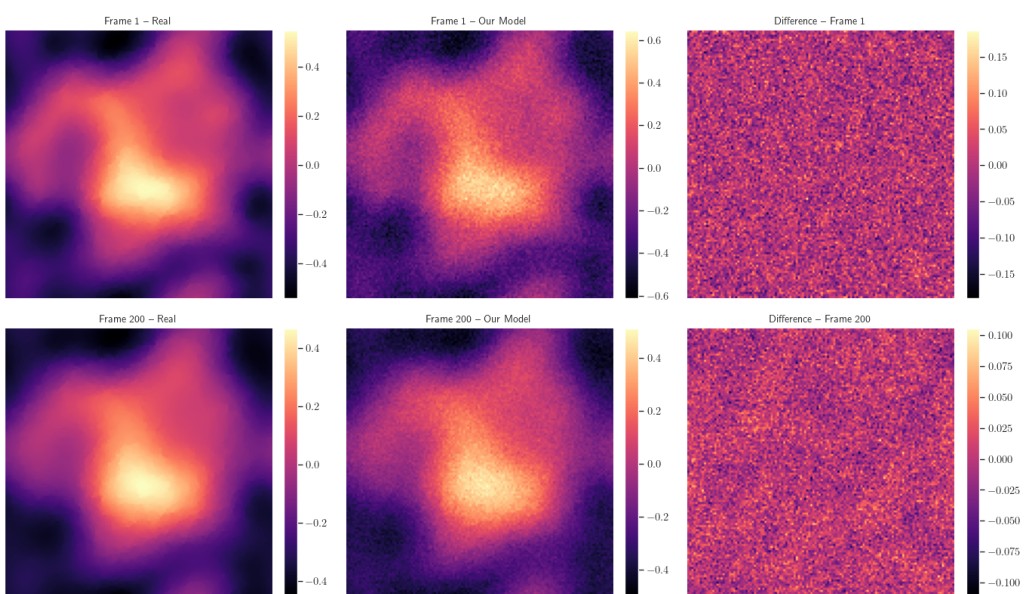

Figure 8: Examples from the heat high-res (test-)dataset, together with the prediction of our model and the difference between the two, at Frame 1 and Frame 200.

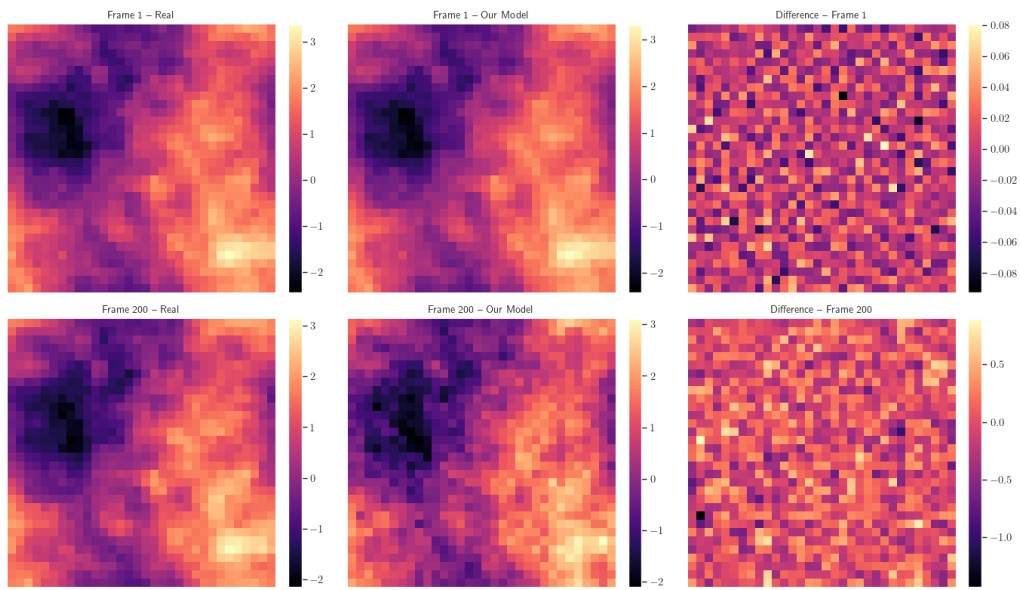

Figure 9: Examples from the wave low-res (test-)dataset, together with the prediction of our model and the difference between the two, at Frame 1 and Frame 200.

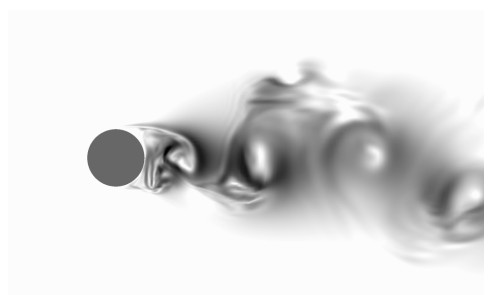
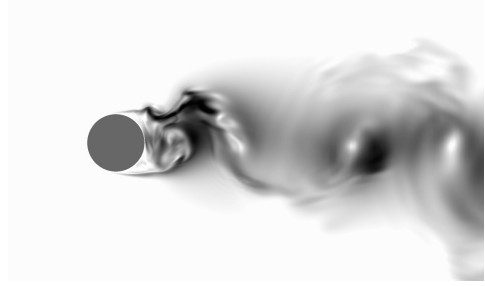

Figure 10: Left: a real example from the LES Karman Vortex street dataset. Right: a still from a video generated by LongVideoGAN.

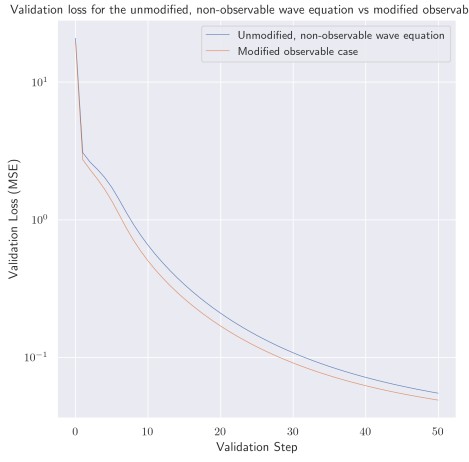
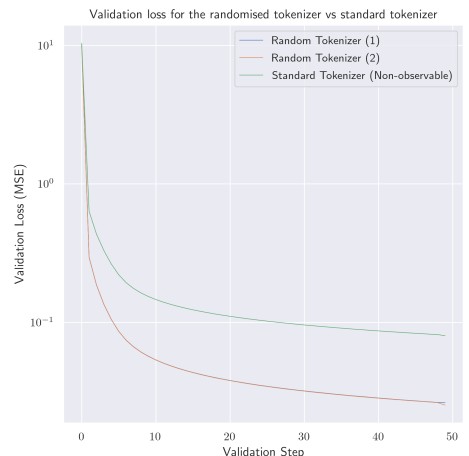

Figure 11: Comparing the validation loss history of the modified, observable wave equation and standard tokenizer pair, with the unobservable, unmodified wave equation and standard tokenizer pair.

Figure 12: Effect of randomising the convolutional filter (tokenizer) on the validation loss, for the heat equation.

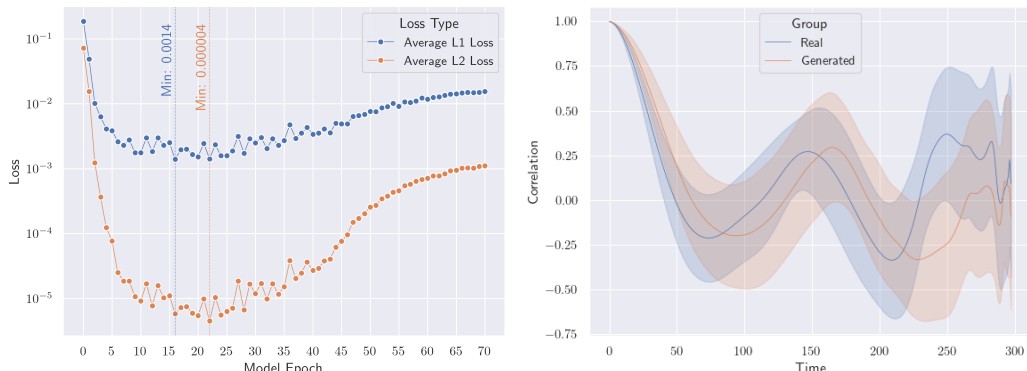

Figure 13: (Left) Residues of our KSE model against epoch, for the low-res test set. For data normalised between $[-1, 1]$, We obtain a minimum average $L_2$ loss of $4 \times 10^{-6}$ and minimum average $L_1$ loss of 0.0014.

Figure 14: (Right) Temporal correlations at the point $(128, 128)$ for the trained model and ground truth. The shaded areas show a $1\sigma$ range. 15 videos were produced as in Fig. 3; the low-res and super-res models are concatenated.

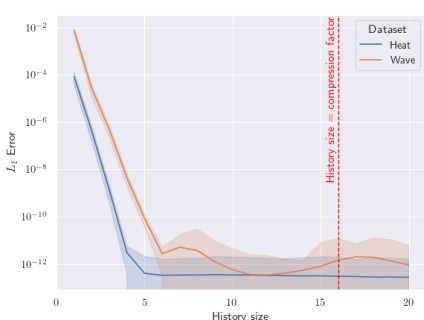

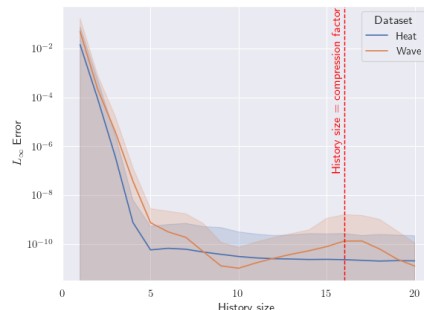

Figure 15: Average $L_1$ error as a function of the model history size, where the model is 'trained' by least-squares regression

Figure 16: $L_\infty$ error as a function of the model history size, where the model is 'trained' by least-squares regression

## K.2 RANDOMISED TOKENIZER

In this section, we randomise the tokenizer and observe the effect on the validation loss, see Fig. 12. We repeated the experiment several times, with different seeds. Instead of using a fixed average pooling layer (downsampling), we changed the parameters of the 4x4 convolutional filter to be normally distributed about 1 (the previous value) with a standard deviation of 0.1. In all runs we found a significant improvement in both error and convergence times, when compared with the unmodified standard tokenizer. For both of these cases, we used the unmodified heat equation, which, together with the standard tokenizer, form a non-observable pair. Note that the randomised tokenizers achieve a much better performance, and, despite one run being slightly better than the other, are bunched together relative to the run with the fixed tokenizer. This further backs up our claim that observability is an important condition for learning, since this is strong numerical evidence that models with observability perform significantly better than those without, as predicted by section 3. It also agrees with the conclusions of Fig. 2, where the history size was varied, and the unmodified heat equation performed much worse than the modified one. Note that a validation step occurs every 2000 epochs.

## K.3 LEAST-SQUARES REGRESSION

Fig. 15 and Fig. 16 show how, when using a least-squares regression model, the history size changes the average $L_1$ and $L_\infty$ errors, together with a 1-$\sigma$ error bar. Details can be found in Section 5.

## K.4 VIDEOGAN GENERALIZATION

To evaluate the generalization capability of the generative model, we compare the generated videos with both our training data and unseen test data. The goal of this is to determine whether the VideoGAN model is actually producing novel outputs that are representative of the overall data distribution, rather than simply replicating the training data. Given a generated video clip of length $n$, we measure its similarity to all possible sub-videos of length $n$ in the real videos. Specifically, for a sub-video $g$ of a generated video, we compute the minimum Euclidean distance to the real data as

$$d(g, V) = \min_{v \in V^n} \|g - v\|_2 \ ,$$

where $V_n$ denotes the set of all consecutive sub-sequences of length $n$ from a video $V$, and $\| \cdot \|_2$ is the Euclidean norm.

This distance $d(g, V)$ quantifies how close the generated clip $g$ is to the closest consecutive sequence in the dataset $V$. We perform this calculation using both the training dataset $V^{\text{train}}$

and the unseen test datasets $V^{\text{Test1}}, V^{\text{Test2}}, V^{\text{Test3}}$ of the same length $V^{\text{train}}$ using subvideos of length 8. The intuition is that a well-generalising model should produce videos where the distance $d(g, V^{\text{train}})$ is comparable to $d(g, V^{\text{test}})$. This would indicate that the model is not overfitting the training data but instead is generating new data that is consistent with the overall data distribution.

To evaluate the generalization capabilities of the LongVideoGAN model, we calculated $d(g, V)$ for 50 different choices of generated 8 frame-length clips $g$ and for all $V \in \{V^{\text{Test}}, V^{\text{Train1}}, V^{\text{Train2}}, V^{\text{Train3}}\}$.

| $V$ | $\min_{g \in G} d(g, V)$ |
|---|---|
| $V^{\text{Train}}$ | 40456.58 |
| $V^{\text{Test1}}$ | 41750.72 |
| $V^{\text{Test2}}$ | 42079.42 |
| $V^{\text{Test3}}$ | 42336.16 |

(a)

| $V$ | $|\{g \in G : V = \arg\min_{v \in \mathcal{V}} d(g, v)\}|$ |
|---|---|
| $V^{\text{Train}}$ | 18 |
| $V^{\text{Test1}}$ | 10 |
| $V^{\text{Test2}}$ | 10 |
| $V^{\text{Test3}}$ | 12 |

(b)

Table 12: (a) The minimum Euclidean distance measured from each of the 4 video sets over all $g \in G$ where $G$ is the set of the 50 clips used for this evaluation. (b) Counts of how often each of the four sets is the closest to a generated clip, where $\mathcal{V} = \{V^{\text{Train}}, V^{\text{Test1}}, V^{\text{Test2}}, V^{\text{Test3}}\}$.

As the tables above show, the smallest distance between any of the generated clips and one of the four video sets was achieved with the training data. Additionally, the training dataset is also the closest match for the largest number of clips in the set $G$. These observations indicate that the generated data is more similar to the training data than to the test data. However, the minimum distances between any of the clips and the various datasets all fall within a 5% range of the minimum distance achieved with the training set. This suggests that the generated data is not simply a near-identical replication of the training data, as we would expect the value of $\min_{g \in G} d(g, V^{\text{Train}})$ to be significantly lower if that were the case. Given that the training data always biases the model towards replicating its distribution, it is unsurprising that the training data is the closest match for 18 of the generated clips. Nonetheless, the fact that 32 of the clips are closer to some of the unseen data further indicates that the model is capable of generalizing effectively and producing videos that represent the real data distribution.

### K.5 Further Temporal Correlations

After analysing the 301-frame videos, we extended the correlation evaluation to 1000-frame videos to assess the model's ability to maintain temporal consistency when generating longer videos. The results show that while the generated 1000-frame videos were still able to replicate the general periodicity and long-term dynamics observed in the real videos, there was a noticeable decline in the accuracy compared to the shorter 301-frame sequences.

For Pixel 1, the 1000-frame videos continued to exhibit periodic behaviour, but the correlations were generally higher than those observed in the real videos for most values of $\Delta t$. The period length appeared shorter on average, leading to extreme values of the mean correlation at different values of $\Delta t$ compared to the real video data and also the 301-frame generated videos. This suggests that while the model retains some level of periodicity, the ability to accurately reproduce this declines when generating longer videos.

Overall, these results suggests that while the model is capable of generating video sequences that capture a lot of the behaviour of the real flow, there is a trade-off between temporal accuracy and generated video length.

## L Navier Stokes Equations for Large Eddy Simulation (LES)

In section 3.3, we introduced the Navier–Stokes equations, on which turbulent flows rely. More specific we obtain our data by Large Eddy Simulation (LES) which is based on the concept of scale separation in turbulent flows, allowing for the large, energy-containing eddies to be resolved explicitly while modelling the effects of the smaller, subgrid-scale motions (Sagaut, 2005). This is achieved through a spatial filtering operation, which separates the

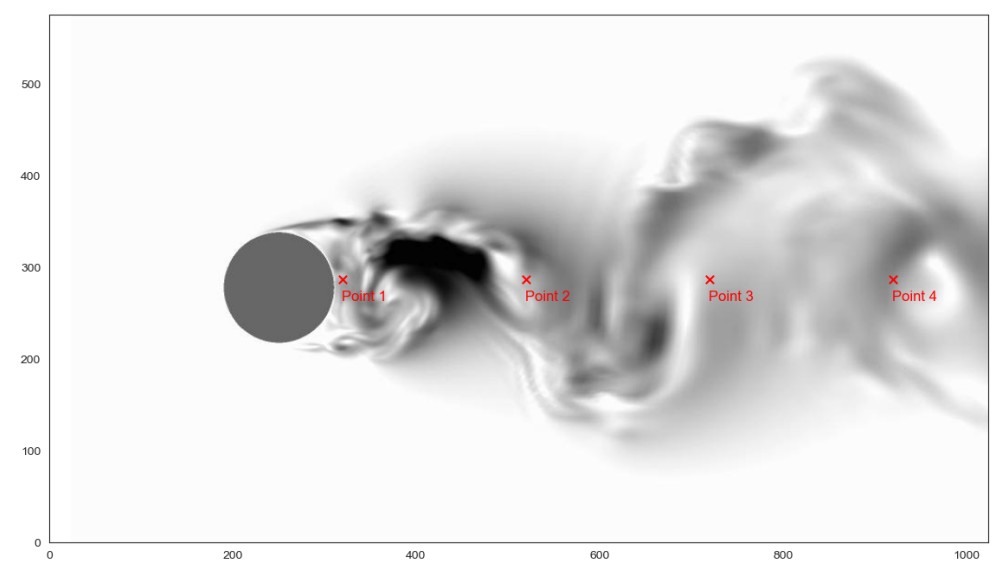

Figure 17: Locations of the pixels used for the temporal correlation analysis using an image from the training data.

flow $\varphi(x,t)$ into a filtered (resolved) component $\overline{\varphi}(x,t)$ and a subgrid-scale component $\dot{\varphi}(x,t)$, such that

$$\varphi(x,t) = \overline{\varphi}(x,t) + \dot{\varphi}(x,t). \tag{24}$$

In implicit LES, the filtered flow $\tilde{\varphi}$ is defined via a convolution integral in physical space:

$$\overline{\varphi}(x,t) = \int\limits_{-\infty}^{\infty} \int\limits_{-\infty}^{\infty} G(x - x', t - t')\varphi(x', t')\, dt'\, dx', \tag{25}$$

where the filter function $G$ is implicitly defined by the numerical discretization, with the filter width corresponding to the local mesh size.

Applying this filtering operation to the NSE, where it is assumed (Sagaut, 2005) that the filtering and differential operators commute, yields the filtered LES form:

$$\nabla \cdot \overline{\mathbf{u}} = 0, \quad \frac{\partial \overline{\mathbf{u}}}{\partial t} + (\overline{\mathbf{u}} \cdot \nabla)\overline{\mathbf{u}} = -\frac{1}{\overline{\rho}}\nabla\overline{p} + \frac{1}{\mathrm{Re}}\nabla^2\overline{\mathbf{u}}, \tag{26}$$

where all quantities are the filtered versions of the original NSE and the Reynolds number is defined as $\mathrm{Re} = \frac{U_\infty D}{\nu}$ with $U_\infty$ denoting the freestream velocity and $D$ the diameter of the cylinder.

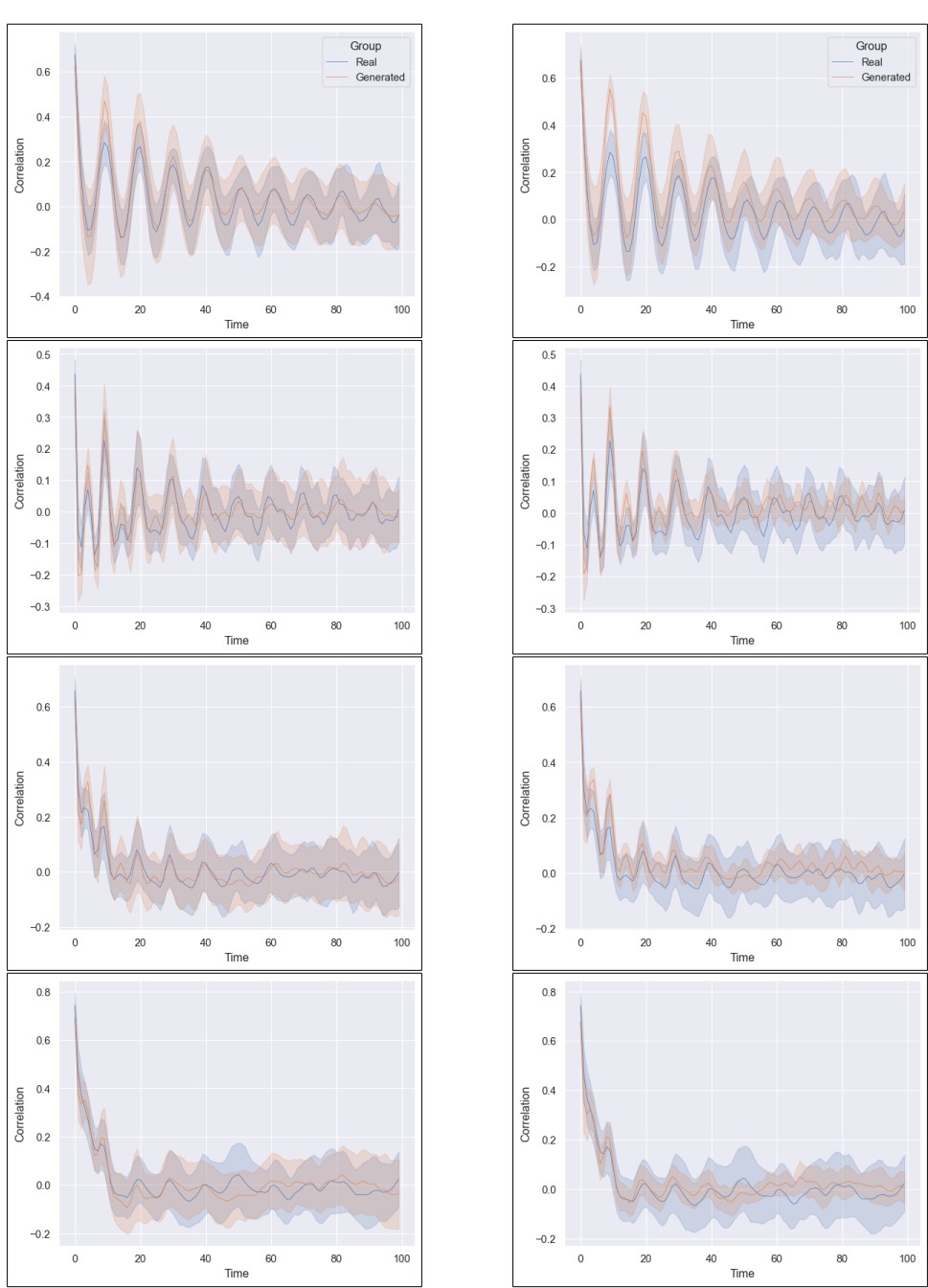

Figure 18: Temporal correlations of generated videos of length 301 (left) and 1000 (right) for pixels 1, 2, 3 and 4 (top to bottom).