# OpenReview forum: "When do World Models Successfully Learn Dynamical Systems?"
_ICLR.cc/2026/Conference — ICLR 2026 Conference Withdrawn Submission_

### Official Review · Reviewer_DW7U · 2025-10-24

**Soundness:** 3
**Presentation:** 3
**Contribution:** 3
**Rating:** 4
**Confidence:** 3

**Summary:**

This paper proposes using control theory concepts, specifically observability, to understand when world models can successfully learn physical dynamical systems. The authors develop a theoretical framework connecting tokenization to observability, then validate it with experiments ranging from simple linear PDEs through chaotic Kuramoto-Sivashinsky equations to turbulent cylinder flow simulations.


The core idea is interesting: they argue that world models work when the underlying system is "observable" - meaning you can reconstruct the full state from a history of tokenized observations. They prove some PAC learning results and demonstrate this with GANs of increasing complexity.

**Strengths:**

- Connecting observability from control theory to world model learnability is genuinely original. I haven't seen this angle before in the ML for physics literature, and it's a useful lens for thinking about why some systems are easier to learn than others. The intuition that you need sufficient observability to learn dynamics makes sense.

- I appreciate the methodical approach - starting with least squares, moving to linear layers, then shallow adversarial networks, finally full GANs. This helps isolate what's driving performance. The choice of numerical methods is also sound (ETDRK4 for KS equation, LES for turbulence).

- The analysis of linear systems using Kalman observability matrices (Theorem 3) was well-executed. The results about the history length needing to exceed the compression factor make sense and are properly validated.

**Weaknesses:**

- Here's my biggest concern: In Theorem 6, the authors prove that standard heat and wave equations with constant coefficients are NOT observable under their patch-averaging tokenisation scheme. The solution? Modify the Laplacian to have position-dependent stochastic coefficients ∇·(a(x)∇u).

If I'm not mistaken, this is an issue. Heat equations ARE observable in the standard control theory sense - you can reconstruct the state from boundary measurements or interior point sensors. The issue is that their specific tokenization loses high-frequency information.

All the linear PDE experiments use these modified equations. This isn't validation - it's engineering problems to fit the theory.

It is interesting in itself, but the paper should discuss those limiations.

- The paper cites FNO and DeepONet extensively in the introduction and related work. But there are ZERO quantitative comparisons in the results. Not one number comparing performance.

- You only use 28 training initial conditions for a chaotic PDE. Isn't it a bit too small?

- I'm less sure about that, but your theoretical framework assumes deterministic reconstruction (continuous map G: Y^k → X). But your experiments use stochastic GANs that learn distributions. GANs optimise distribution matching, not pointwise reconstruction. They can have mode collapse, generate diverse samples, etc.
There's a fundamental mismatch between your deterministic theory and your stochastic experiments. The GAN's success doesn't necessarily validate your observability framework?

**Questions:**

- What happens if you test on unmodified heat equations?
- How does your method compare quantitatively to FNO and DeepONet?
- How do you resolve the mismatch between deterministic theory and stochastic GANs?

---

> ### Author Response · Authors · 2025-11-21
>
> Dear Reviewer DW7U,
>
> We would like to thank you for your time and constructive review of our work. We especially appreciate the attention you have paid to the finer details of our manuscript. We would like to respond to a few of the points you have raised.
>
> 1. We are not quite sure why modifying the Laplacian is a problem here. You state that it’s an issue, but not why. The purpose is to contrast the observable with the non-observable case, given in fig. 2. The results exactly match what we expect given the theory. That the random perturbation leads to an observable system with probability one, is an easy consequence from applying the perturbations to the observability matrix (eq. 6), since the perturbations are continuously distributed; hence, with probability one, the determinant of this matrix is nonzero and the matrix has full rank. Of course, equally, instead of modifying the dynamics, we could modify the downsampling convolution. To this end, we repeated our experiments for a randomised tokenizer. The results will be included in a follow-up comment in the coming few days due to space constraints in this rebuttal.
> 2. The claim that there are ‘ZERO’ quantitative comparisons to FNOs and DeepONets is abjectly false. We refer you to tables 1, 2 and 3, with table 1 in the main text, and tables 2 and 3 referred to in the main text on lines 407-408 and 406 respectively.
> 3. Using 28 initial conditions is, to our knowledge, fine given our time horizon of 2000 frames + 500 discarded burn-in frames, with every frame representing 10 timesteps in the ODE solver. Whilst there is no formal result on the ergodicity (which is enough, see [1]) of the deterministic KS equation, there are results on the global attracting set [2] and the exponential mixing (and ergodicity) of the stochastically modified version [3][4], and the equation is certainly chaotic, so we believe our setup is sufficient.
> 4. For adversarial training, a probabilistic framework is not needed. Lines 239-240 in the main text point the reader to Appendix D where this is explained and proven. We employed this ‘deterministic’ GAN training for all of our models with the exception of the LES-based experiments in CFD, where we train based on the Jensen-Shannon divergence. In this case, this is justified, because we work with data from a partially observable state (the turbulence strength) and not the full flow field. Therefore in the last experiment, the application of the probabilistic framework is justified.
>
> ## Questions
>
> 1. This question is explored and answered in lines 359-266, with particular reference to fig. 2. Above, in point 1, we answer the question of instead modifying the tokenizer.
> 2. This question is answered above, in point 2.
> 3. This question is answered above, in point 4.
>
> Many thanks for your constructive review. We hope we have answered your questions to a sufficient standard. If you have any further concerns, please do let us know.
>
> Many thanks,
>
> The Authors
>
> ## References
> [1] Drygala et. al., 2022, Generative Modeling of Turbulence, https://arxiv.org/pdf/2112.02548
>
> [2] Collet et. al., 1993, A Global Attracting Set for the Kuramoto-Sivashinsky Equation
>
> [3] Gao & Nguyen, 2025, Exponential mixing for the stochastic Kuramoto-Sivashinsky equation on the 1D torus, https://arxiv.org/abs/2508.01794
>
> [4] Weinan & Liu, 2002, Gibbsian Dynamics and Invariant Measures for Stochastic Dissipative PDEs, https://doi.org/10.1023/A:1019747716056

---

### Official Review · Reviewer_yVYL · 2025-10-25

**Soundness:** 3
**Presentation:** 3
**Contribution:** 2
**Rating:** 4
**Confidence:** 2

**Summary:**

This paper provides a theoretical framework to study "when do `world models` successfully simulate physical systems", using approaches from control theory. It defined the notion of "observability" of a system, and ties it to its learnability-- namely the ability to recover the full state c from a history of low-dimensional "tokenized" observations. Experiments on linear PDEs, chaotic equations, and turbulent flows validate this, showing the model outperforms neural operators like FNO and DeepONet in long-term stability.

**Strengths:**

- **Originality**: The paper's main strength is in merging concepts from diverse fields -- linking concepts from control theory, "observability" to generative "world models" using , and further framing the latter as operator learning. In doing so it provides a framework that allows deeper insight on the expected performance of world models.

- **Quality**: The work is of good quality, providing rigorous theoretical grounding alongside empirical validation. The theory is formally presented with clear definitions and theorems, with proofs available in the appendix. The experiments span diverse linear and non-linear systems. The quantitative evaluation support claims on performance and robustness.

- **Clarity**: The paper is well-written and logically structured. Providing the necessary background, definitions and theoretical foundation, towards the experimental evaluation.

- **Significance**: The impact of this work is in providing a  theoretical handle to analyze world models-- guidance when would generative models for physical simulation be accurate.

**Weaknesses:**

- **Assumptions / scope of theory**: The main analysis relies on "global observability" and the existence of a continuous inverse
map $G$. A strong condition; which as indicated by the authors in nonlinear PDEs it is rarely checkable, following this the PAC theorems are qualitative.
- **Circular validation**: Following the previous comment, the authors specifically state that observability for the full turbulent flow is unknown and thus rely on the model's success as "experimental evidence". This creates a circular loop--the theory is supposed to explain success, but success is used to infer back the theory (the observability).
- **Broader Impact / Applications**: While the general appeal of the approach is clear the paper is lacking a discussion or presentation of possible applications or use cases beyond accuracy measurements.

**Questions:**

The following questions are wrt to the above weaknesses:
- **Assessing observability**: The key contribution is the link between observability and model performance. However, proving observability for complex non-linear systems is often intractable. To avoid the "circular" validation would it be possible to derive diagnostics to estimate whether a system is "observable enough"?
- **Broader impact**:
(1) Beyond the provided metrics, showcasing improved performance, could you elaborate on specific, practical applications where the framework could provide a unique advantage / contribution.
(2) Could you elaborate more broadly on the contribution and significance of this work.

---

> ### Author Response · Authors · 2025-11-21
>
> Dear Reviewer yVYL,
>
> We would like to thank you for your time and constructive review of our work. We especially appreciate the attention you have paid to the finer details of our manuscript. We would like to respond to a few of the points you have raised.
>
> 1. Indeed, the existence of the continuous inverse map $G$ follows directly from the observability of the system and the inverse function theorem, and therefore also on the continuity of the forward dynamics. We provide a diagnostic of local observability of the KS equation in Appendix F (and in particular, fig. 7). The difficulty arises from calculating higher dimensional Lie derivatives, which becomes increasingly intractable. However, cutting off these derivatives at an appropriate point and evaluating the rank of the observability matrix can show if the dynamics are not observable, and increases confidence in observability when the matrix has full rank. One other possible solution would be resorting to computer algebra methods. This means the PAC theorems are not purely qualitative. Despite PAC-conditions only giving worst case estimates, they clarify when successful learning is feasible and when it's not. They are thus a valuable tool to analyse failure. Take the non-observable heat equation, where the conditions of our PAC framework explain the numerical effects observed.
>
> 2. We do not believe this is a circular argument, since we don’t make a concrete claim that observability must hold due to the success of the models. Rather, we try to build a picture, starting with examples where the observability is known, demonstrating the expected behaviour, and then moving to examples where it is unknown. In machine learning, conditions for theoretical guarantees are rarely met in the real world, and thus we can only prove a strict relationship between observability and reconstruction of the state in simple cases. For non-toy examples, the relationship is trickier.
>
> 3. There are a number of potential use cases for this work. To begin with, we show an improvement over existing models, and the use case for modelling dynamical systems is vast. It is applicable in places where integrating a complex underlying PDE is impractical or unfeasible, and only a subset of datapoints are available. The uses for this include uncertainty quantification, parameter estimation, surrogate modeling for control and optimisation purposes, and so on. Downstream tasks include things like digital twins [1,2,3], real-time and predictive control [4,5], solving inverse problems [6,7] and augmenting datasets/scenario generation [8,9] but to name a few.
>
> ## Questions
>
> 1. We answer this question above in point 1, and point to a diagnostic for observability.
> 2. We also give some examples of use cases in point 3.
> 3. Certainly. We try to restate the main contributions clearly:
> We are the first to (a) isolate the ‘autoregressive with two-resolution models’ approach and identify and interpret it as a ‘world model’ applied to dynamical systems/operator learning, then compare to the existing state-of-the-art methods such as FNO and DeepONet; (b) find a significant performance increase for long-term horizons over these models; and (c) apply a system-theoretic PAC framework to show the necessity of observability in learning these systems, as an ingredient of the success of applying world models in these scenarios.
>
> Many thanks for your constructive review. We hope we have answered your questions to a sufficient standard. If you have any further concerns, please do let us know.
>
> Many thanks,
>
> The Authors
>
> ## References
>
> [1] G. Tsialiamanis, et al. "On generative models as the basis for digital twins." Data-Centric Engineering 2 (2021): e11.
>
> [2] Chakraborty, et. al. "The role of surrogate models in the development of digital twins of dynamic systems." Applied Mathematical Modelling 90 (2021): 662-681.
>
> [3] Y.-C. Lai. "Digital twins of nonlinear dynamical systems: A perspective." The European Physical Journal Special Topics 233.6 (2024): 1391-1399.
>
> [4] L. Yang, et. al. "Physics-informed generative adversarial networks for stochastic differential equations." SIAM Journal on Scientific Computing 42.1 (2020): A292-A317.
>
> [5] M. Celia, et. al. "A comparative study of neural ordinary differential equations and neural operators for modeling temporal dynamics." Neural Computing and Applications 37.30 (2025): 25319-25338.
>
> [6] O. Ernst, et al. "Learning to Integrate." arXiv preprint arXiv:2506.11801 (2025).
>
> [7] P. Krueger, et. al.. "Generative design of a gas turbine combustor using invertible neural networks." Journal of Engineering for Gas Turbines and Power 147.1 (2025): 011007.
>
> [8] E. Parish, et. al. "A paradigm for data-driven predictive modeling using field inversion and machine learning." Journal of computational physics 305 (2016): 758-774.
>
> [9] A. Singh, et. al. "Machine-learning-augmented predictive modeling of turbulent separated flows over airfoils." AIAA journal 55.7 (2017): 2215-2227.

---

### Official Review · Reviewer_36Bq · 2025-10-31

**Soundness:** 4
**Presentation:** 3
**Contribution:** 2
**Rating:** 6
**Confidence:** 4

**Summary:**

The authors investigate when world models, i.e. latent generative models with autoregressive temporal dynamics, can successfully learn the behavior of dynamical systems. They frame these models within a control-theoretic context, so that observability determines whether the true system dynamics can be recovered from tokenized latent representations. Using both a novel PAC theoretical apparatus and a series of experiments, they show that a finite-history autoregressive map exists and can be learned only if the system is observable from its latent outputs. Empirical tests on linear and nonlinear PDEs, including the heat, wave, and Kuramoto–Sivashinsky equations, demonstrate that increasing the autoregressive history length improves prediction accuracy up to the point where observability is achieved. The authors further show that their world-model framework, implemented using a hierarchy of increasingly complex architectures culminating in video GAN models, produces long-horizon stable simulations of fluid and chaotic systems and outperforms neural operator baselines such as FNO and DeepONet in stability and efficiency.

**Strengths:**

1. To my knowledge, the theoretical PAC formulation of these latent, autoegressive models is new contribution to data-driven dynamical systems. The authors proceed very systematically and the rigor and clarity of their theoretical framework and experimental results is well-received. The empirical predictions the authors make about when models will be easily learned or not are strong and elegant.

2. Benchmarking against other methods in terms of performance and complexity is very thorough.

**Weaknesses:**

1. The principal weakness of this paper is its rhetorical presentation. In particular, introduction makes it very hard to understand what problem the authors want to solve and how their approach differs from other methods. After reading, it becomes clear that they seek to address the lack of applications of auto-regressive, latent-space models (i.e. world models) to physical systems rather than video generation, particularly with rigorous guarantees of performance based on observability. Furthermore, the authors refer mid-way through to "our model", but it is unclear at this point what the "model" is since all architectural details are relegated to the (extremely detailed) appendix. It seems as if what the authors refer to as their "model" is indeed the autoregressive approach itself, independent of the implementing architecture, which varies by application. If this is indeed their contribution, it must be stated much more clearly. It took me a while to realize in retrospect that, indeed, most forecasting models for dynamical systems take a one-step approach, predicting x_t+1 from x_t with a detail through latent space (Champion et al., 2019 PNAS; Vlachas et al. 2022 NMI ). Again, this is a rhetorical weakness, but it has the result of destabilizing and disorienting the reader. The motivation and material contribution must be stated exactly.

2. Comparison to previous world learning/autoregressive models for physical systems is weak. The authors mention work by Skorokhodov and Klemmer and say that "they neither analyse system-theoretic foundations nor compare with operator-learning approaches". But why is that a fatal flaw? In that sense, are you arguing that your main contribution is the PAC theoretical work and operator learning benchmarking? Or do you offer a performance boost as well? Also, from some brief googling, I found: (1) https://www.nature.com/articles/s41598-024-68944-0? and (2) https://eurasip.org/Proceedings/Eusipco/Eusipco2022/pdfs/0002216.pdf?. These both seem like autoregressive models for physical systems. I'm not saying you have to compare directly to these, but rather that I, as a reader, was left a bit confused as to what was novel between your approach and these earlier methods.

In summary, this is a technical tour de force, but its concrete novelty and "problematic" must be emphasized.

**Questions:**

Summarizing based on the weakness section:

1. What are your concrete contributions and what problem are you solving?

2. Assuming those contributions are not 100% novel, how do they differ from previous approaches? For example, you cite a lot of interesting work in the Related Work section, but nowhere do you say how those approaches differ from or are weaker than what you propose. This makes the paper seem like a very dry, technical report as oppose to a novel contribution.

---

> ### Author Response · Authors · 2025-11-21
>
> Dear Reviewer 36Bq,
>
> We would like to thank you for your time and constructive review of our work. We especially appreciate the attention you have paid to the finer details of our manuscript. We would like to respond to a few of the points you have raised.
>
> 1. We agree that the introduction would benefit from a stronger system-theoretic framing, and that the use of the phrase ‘our model’ is confusing given that, as you correctly point out, it refers more to an approach than a particular concrete model. The revised version discusses the state-space interpretation, observability considerations, and connections to operator learning, and clarifies their relevance to our contributions. The revised version is included as an additional comment below this response. We will also modify the use of the phrase ‘our model’ elsewhere in the paper.
>
> 2. Thank you for these interesting references, we will integrate them into the main paper. They appear to aim to solve specific dynamical problems with specific autoregressive model architectures. You are correct to identify that we are not the first to apply an autoregressive model to dynamical systems. However, we are the first to (a) isolate this approach and identify and interpret it as a ‘world model’ applied to dynamical systems/operator learning, then compare to the existing state-of-the-art methods such as FNO and DeepONet; (b) find a significant performance increase for long-term horizons over these models; and (c) apply a system-theoretic PAC framework to show the necessity of observability in learning these systems, as an ingredient of the success of applying world models in these scenarios.
>
> World models use the latent space to learn the global dynamics in a more stable latent/reduced order representation. This is important for long-term stability, as we have shown empirically. Mathematically, this is reflected by the fact that there are lesser parameters to be learned as we only require a $n \times (n/q)$ matrix in the linear case.
>
> ## Questions
>
> 1. Is answered above.
> 2. We will add comments to the related works section, emphasising what we contribute over the cited works.
>
> Many thanks for your constructive review. We hope we have answered your questions to a sufficient standard. If you have any further concerns, please do let us know.
>
> Many thanks,
>
> The Authors

---

> > ### Author Response · Authors · 2025-11-21
> >
> > ## New introduction
> >
> > World models were introduced by Ha et. al. as generative models in which a high-dimensional state of the environment is compressed into a latent representation whose dynamics are learned autoregressively. As emphasized by Lecun, such latent dynamics should ideally encode the physical laws governing the environment rather than merely reproducing observational correlations. Interpreting world models in this way naturally connects them to state-space modelling, observability, and operator learning. Recent advances follow this paradigm by using powerful autoregressive sequence models to capture temporal structure in the latent space, combined with super-resolution modules that reconstruct high-resolution states. This approach has produced stunning results in video generation and dynamic scene synthesis (Hu et. al., Peng et. al.).
> >
> > Although these developments are often described within the context of generative modelling, world models can be understood more fundamentally as learning dynamical systems. The encoder acts as an observation operator that compresses a short temporal window of the full state into a low-dimensional representation, while the latent dynamics module predicts the next latent state based on this compressed history. A reconstruction module then lifts the latent representation back to the full-dimensional state. Interpreted system-theoretically, this architecture defines a learned reduced-order state-space model. Its effectiveness depends crucially on whether the compressed representation preserves information necessary for identifying the underlying physical state and whether the learned latent dynamics provide a stable approximation of the true evolution of the system.
> >
> > This system-theoretic view also reveals a natural conceptual connection to neural operator learning, where the goal is to learn the evolution of physical systems from data and efficiently generate solutions to systems governed by partial differential equations (Li et al., Lu et al., Kovachki et al., Zhang et al.). Neural operators are trained on sample trajectories produced by numerical simulation and then generalise to new initial conditions, from which full solutions to the physical equations can be recovered.
> >
> > Both world models and neural operators aim to approximate the underlying dynamical operator of the system, but they do so in fundamentally different ways and under different assumptions, giving rise to several open research questions. These include how observability is maintained under compressed latent representations, how well autoregressive latent dynamics approximate true PDE evolution operators, and how the length of the latent-history window relates to the effective dynamical order of the underlying system.
> >
> > Generative modelling of physical systems has already been widely explored with an emphasis on turbulent flows, see e.g. Kim et. al. and Drygala et. al.. A few works have also explored spatio-temporal generative modelling of the type discussed in Section 1. However, these works rely primarily on video similarity metrics and do not provide theoretical or empirical studies of the physical accuracy of the generated data. World models have also been applied to simulate physical systems (Skorokhodov et al., Klemmer et al.), but these efforts focus on video generation and therefore neither analyse the system-theoretic foundations nor compare their performance to operator-learning approaches, leaving the research questions stated above open.
> >
> > Our contributions in this paper are as follows:
> >
> > 1. We introduce a system-theoretic framework to understand when world models can reliably learn dynamics from tokenized observations. This is demonstrated through experiments on a hierarchy of dynamical systems, starting from the heat and wave equations to complex turbulent flow synthesis. The study is mathematically grounded, explaining operator learning within world models, and proving that neural networks struggle to capture the time-evolution operator in systems that lack observability. Video visualisations of our solutions can be found in the supplementary material.
> > 2. We interpret world models as an operator learning method and compare this approach to state of the art methods like FNO and DeepONet, showing superior long-term stability of world models for in-distribution and out-of-distribution test data.
> > 3. We provide numerical studies on the length of token history in autoregressive predictions and super-resolution reconstruction, which is a first study of autoregressive model order reduction for dynamical systems.
> > 4. We show that world-model based simulation can achieve low error predictions in capturing the key temporal correlations in turbulence modelling, even in situations where the state cannot be fully observed, at significantly less computational cost than traditional methods from computational fluid dynamics.

---

### Official Review · Reviewer_1aew · 2025-11-01

**Soundness:** 2
**Presentation:** 2
**Contribution:** 2
**Rating:** 2
**Confidence:** 3

**Summary:**

This paper proposes to study when world models can successfully learn dynamical systems. The authors claim a system-theoretic and PAC-learning framework linking observability to learnability and present numerical experiments on simple PDEs (heat, wave, KSE) and a CFD case (vortex shedding).

**Strengths:**

The paper raises an interesting conceptual question connecting world models and operator learning.

The experiments cover several standard PDE benchmarks and attempt to bridge control theory and data-driven modeling.

**Weaknesses:**

1. The paper does not answer “when world models succeed” in any rigorous sense. The so-called theoretical results restate standard observability conditions and a generic PAC-learning existence theorem, without offering new criteria or quantitative conditions. The main theorems are textbook results in nonlinear systems (Kalman, Hermann–Krener) and elementary PAC learnability statements.

2. It is not clear why the models used are “world models.” They are standard autoregressive video predictors applied to PDE snapshots,

3. The “observability” discussion is never quantitatively linked to training outcomes; the perturbation of coefficients in the heat equation cannot be interpreted as testing observability. The theoretical part is largely decorative.

4. The paper does not clarify what fundamentally distinguishes its proposed model from standard neural operator methods such as FNO or DeepONet. From the reviewer’s perspective, there are at most two apparent differences: 1. It applies an autoencoder to learn latent dynamics. 2. It includes memory by feeding the entire trajectory history instead of the final frame.
However, both aspects can be trivially incorporated into baseline operator-learning models, and the paper does not explain why these would yield fundamentally different behavior. Moreover, the implementation details of FNO and DeepONet are not described, and the extent of ablation studies is unclear. Hence, it is difficult to assess whether the observed improvements are due to architectural choices or merely training conditions.

5. The numerical comparisons with FNO and DeepONet are limited to table-level metrics without controlling for architecture size, data resolution, or rollout time. Claims of “superior long-term stability” lack statistical justification, and no variance or ablation results are provided.

**Questions:**

1. The presentation of Definition 1 is unclear. Specifically, the function \( y \) appears to depend only on \( t \) in equation (2), whereas in Definition 1 it is defined as a function of both \( t \) and \( x \). This discrepancy should be clarified to avoid confusion regarding the role of each variable.

2. The definition of G is missing. Since G appears to play a role in the formulation or analysis, its precise meaning and mathematical structure should be clearly stated to ensure the reader can follow the development and verify the results.

3. The manuscript should demonstrate that the set of defined observable pairs is nonempty. Without such a justification, it remains unclear whether the proposed framework admits any valid instances.

4. The study is presented as theoretical but is mainly empirical; it does not develop new algorithms or theory, nor does it analyze failure modes beyond trivial observability remarks.

---

> ### Author Response · Authors · 2025-11-21
>
> Dear Reviewer 1aew,
>
> We would like to thank you for your time and constructive review of our work. We especially appreciate the attention you have paid to the finer details of our manuscript. We would like to respond to a few of the points you have raised.
>
> 1. We strongly disagree. Our central theoretical result (theorems 4 and 5) is to introduce the notion of observability into the discussion of world models - or autoregressive models on a latent space - learning dynamical systems. The novelty of this result is not disputed in any of the referee reports. To our knowledge, the connection between observability and PAC learning in this area was not already known. If you disregard the central idea of a paper, most papers become trivial. If you are able to find a reference to the contrary, please provide it, and we will gladly revise this point. Theorem 3 is a standard result, readily found in textbooks; we have never presented it as our own result, and we cite it appropriately. Concerning the quantitative rates, yes, we could turn on the SLT-machine, combine chaining and concentration inequalities with the latest results on approximation rates for DNNs, e.g. by [1]. Instead, we focused on novelty and providing numerical examples, deciding this was more effective than further extending the appendix. Theorem 4 establishes learning based on the existence of a solution to the autoregressive problem as a map. Theorem 5 shows PAC learning for a non-trivial two level problem where one has to account for the errors of the lower level autoregressive system to prove PAC-learning.
>
> 2. We are a little surprised by this comment. We used the terminology that is commonly used for exactly these kinds of models in the scientific community. In our eyes, this is a matter of respect and giving credit, and is not a weakness.
>
> 3. We respectfully disagree. As explained in lines 273-278, in figure 2 we see a drop in learning performance/rise of residual error due to a non observable dynamical system. In contrast, not only do the perturbed heat equations have a lower error when trained with sufficient history length, but the error does not improve when increasing the history length beyond this threshold. Hence there is a theoretically predicted effect that is observed in training and the theory cannot be deemed merely decorative. That the random perturbation leads to an observable system with probability one, is an easy consequence from applying the perturbations to the observability matrix (eq. 6), since the perturbations are continuously distributed; hence, with probability one, the matrix has full rank. We will also provide a further experiment, where we regain observability of the heat equation by instead perturbing the tokenizer. Due to space constraints in this rebuttal, the results will be included as a follow-up comment. We will present the results of the experiment in the appendix and briefly refer to the above argument in the main text.
>
> 4. Thank you for this remark; we provide details on these models in point 5. Is it presumably possible to incorporate the two key features of world models - the two-resolution model and the autoregressive history - into the FNO and DeepONet approaches, but then these architectures would have changed considerably, and resemble our approach. This could be a point for future work.
>
> 5. Thank you for this point, we aim to rectify it here. The FNO used in the paper is developed by PhysicsNemo (formerly Modulus), a package developed by NVIDIA. The hyperparameters can be found in Table 1 below.
>
> Below you can find Table 2 comparing the Euclidean distance between the correogram of the model and the data for the KS equation, as done in the main paper (table 2). In total, we trained 144 additional models, of which we picked the best 10, discriminated by performance on the validation set. Models above a threshold validation loss at epoch 25 were stopped early. Whilst we could explore the parameter space further, or repeat for the DeepONet, there sadly isn’t time in this rebuttal period, but would be included at the earliest available opportunity.
>
> Models are written in the form (no. of layers, decoder layer size, Fourier modes, hidden channels). The results of the ablation study show that a somewhat smaller model than the one we used is more effective, with the best at $t=200$ being (3, 16, 24, 32) and at $t=400$ being (2, 16, 12, 32), other than our model. This table reinforces our claim that our approach is much more effective for longer time horizons. We also include Table 3 below which demonstrates the searched parameter space.
>
> The DeepONet consists of the same FNO architecture as the branch net, and a fully connected network as the trunk. Here, the FCN is built by us and connected to the branch net in the usual manner. The trunk net has 3 layers and 150 hidden channels.
>
> Further details can be found in the code, given in the supplementary material. This rebuttal is continued in the comment below.

---

> > ### Author Response · Authors · 2025-11-21
> >
> > ## Questions
> > 1. We follow the standard mathematical conventions that given a function $u(t,x)$, $u(t)$ stands for the function in $x$ at time $t$, $u(t): x \rightarrow u(t,x)$. Note that in our context, the function is mostly discretised in $x$, so $u(t)$ can equivalently be seen as a vector in the dimension of the discretisation. Although this is all standard, we can make this explicit to make the text easier for the non math-oriented reader.
> >
> > 2. This is a good point. The definition is hidden in lines 153-155 where G is introduced with reference to Figure 1. We will provide a more formal definition.
> >
> > 3. Thank you for this comment. We will include a remark in the main paper. For the explanation, we refer to the above discussion which proves that with probability one the perturbed system - both if you perturb the dynamics or the tokenizer randomly  - is observable. For the KS equation (in 1+1 dimensions), we refer to the evidence for local observability after transition to chaos given in Appendix F, which we referred to in lines 285-286 in the main paper.
> >
> > 4. This statement appears to be a rehash of earlier comments made in the weaknesses section.
> >
> > We believe we have sufficiently addressed the points made in your review. If you have any further questions or concerns, we would be happy to hear them.
> >
> > Many thanks,
> >
> > The Authors
> >
> > ## References
> >
> > [1] Belomestny, Denis, et al. "Simultaneous approximation of a smooth function and its derivatives by deep neural networks with piecewise-polynomial activations." Neural Networks 161 (2023): 242-253.
> >
> > ## Tables
> >
> > ### Table 1
> >
> > - Decoder layer size: 32
> > - Hidden channels: 64
> > - Fourier modes: 24
> > - No. of Layers: 4
> > - Learning rate: 0.001
> > - Max epoch: 100
> >
> > ### Table 2
> >
> > | Model              | Correlation at $t=50$    | Correlation at $t=200$   | Correlation at $t=400$   |
> > |--------------------|---------|---------|---------|
> > | Ours               | 2.3749  | 5.6150  | 11.9308 |
> > | $(2, 8, 12, 16)$   | 1.7484  | 6.3689  | 14.6852 |
> > | $(2, 8, 24, 16)$   | 1.6072  | 6.2392  | 14.8404 |
> > | $(2, 8, 12, 32)$   | 1.5180  | 6.3100  | 14.9246 |
> > | $(2, 8, 24, 32)$   | 1.6583  | 6.4774  | 14.8132 |
> > | $(3, 16, 24, 32)$  | 1.5537  | 5.6931  | 14.5094 |
> > | $(3, 16, 24, 16)$  | 1.5214  | 6.2221  | 14.8424 |
> > | $(2, 16, 24, 16)$  | 1.5779  | 5.6759  | 14.7714 |
> > | $(2, 16, 12, 32)$  | 1.8459  | 6.1103  | 14.4958 |
> > | $(2, 16, 24, 32)$  | 1.6168  | 6.3857  | 15.3323 |
> > | $(3, 16, 12, 32)$  | 1.5153  | 6.1225  | 14.7511 |
> >
> > ### Table 3
> >
> > | Setting              | Values                           |
> > |----------------------|-----------------------------------|
> > | No. of layers        | $[1, 2, 3, 4, 5]$                   |
> > | Decoder layer size   | $[8, 16, 32, 64]$ (if 1 layer use upper half; if 5 layers use lower half of this range) |
> > | Fourier modes        | $[12, 24, 48]$                      |
> > | Hidden channels      | $[16, 32, 64]$                      |

---

> > > ### Comment · Reviewer_1aew · 2025-11-25
> > > **Thanks for the response**
> > >
> > > Thanks for the response and the additional numerical results. Yet, some aspects remain unclear to me.
> > >
> > >  1. “the theoretical result (Theorems 4 and 5) is to introduce the notion of observability into the discussion of world models.”
> > > However, the link between Theorems 4 and 5 and the observability is still not apparent. It is not clear in what precise sense these theorems formalize observability.
> > >
> > > 2. In addition, the reply seems to suggest that observability is the key reason behind the conclusions in “When do World Models Successfully Learn Dynamical Systems?” Am I understanding this correctly? However, the original abstract and the list of main contributions in the introduction do not appear to emphasize such a point, nor does any theorem explicitly state such a conclusion, which makes it difficult to judge how essential observability is within the overall framework.
> > >
> > > 3. If observability is meant to play a primary role, I would also recommend adding a small summary table at the beginning of the numerical results section indicating which examples are observable and which are not.
> > >
> > > 4. After rereading the paper, my understanding is that function y(t, x_0) does not represent the value of the function at the initial state x_0 evaluated at time (t). Rather, it appears to denote the function evaluated at the state x(t) reached from the initial condition x_0.

---

### Author Response · Authors · 2025-12-01
**Experiment with randomised Tokenizer and nonobservable**

In this comment we provide the results of our experiment with a randomised tokenizer. We repeated the experiment several times, with different seeds. Instead of using a fixed average pooling layer (downsampling), we changed the parameters of the 4x4 filter to be normally distributed about 1 (the previous value) with a standard deviation of 0.1. In all runs we found a significant improvement in both error and convergence times, when compared with the unmodified standard tokenizer. For both of these cases, we used the unmodified heat equation, which, together with the standard tokenizer, form a non-observable pair.


| Model | MSE Validation Loss at 0th Step | 5th | 10th | 15th | 20th | 25th | 30th | 35th | 40th | 45th |
|---|---|---|---|---|---|---|---|---|---|---|
| Randomised tokenizer (Run 1) | 10.178943480000 | 0.086027674710 | 0.053518912210 | 0.043424760930 | 0.038011704500 | 0.034488815860 | 0.031956113220 | 0.029972901520 | 0.028395119060 | 0.027086200260 |
| Randomised tokenizer (Run 2) | **10.178541110000** | **0.085868438370** | **0.053412676230** | **0.043312045810** | **0.037905072870** | **0.034395145130** | **0.031856508220** | **0.029877377540** | **0.028311136610** | **0.027002904880** |
| Non-observable, fixed tokenizer | 10.363128960000 | 0.220287117000 | 0.146004260800 | 0.123086117500 | 0.110453135700 | 0.101981785200 | 0.095680288520 | 0.090744439120 | 0.086671227480 | 0.083280041980 |

Note that the randomised tokenizers achieve a much better performance, and, despite one run being better than the other, are bunched together relative to the run with the fixed tokenizer. This further backs up our claim that observability is an important condition for learning, since this is strong numerical evidence that models with observability perform significantly better than those without, as predicted by our theory section in the main paper. It also agrees with the conclusions of Fig. 2, where the history size was varied, and the unmodified heat equation performed much worse than the modified one. Of course, graphing these results is a better choice for this data, and we will add such a graph to the updated paper. Note that a validation step occurs every 2000 epochs.

In order to further back up this point, we also trained on the non-observable (unmodified) wave equation and standard tokenizer pair. All other parameters remained the same. As the table demonstrates, we again obtain significantly reduced performance when compared to the observable (modified equation) case.


| Model | MSE Test Loss at 0th Step | 5th | 10th | 15th | 20th | 25th | 30th | 35th | 40th | 45th |
|---|---|---|---|---|---|---|---|---|---|---|
| Wave Equation (Unmodified) | 20.83079041 | 1.716902758 | 0.651929798 | 0.3381791506 | 0.2089028249 | 0.1441555506 | 0.1078062717 | 0.08569634278 | 0.07141580474 | 0.06169737671 |
| Wave Equation (Modified) | **19.59404609** | **1.368278172** | **0.501506865** | **0.2659195404** | **0.1685876579** | **0.1190948206** | **0.09085001445** | **0.07348476558** | **0.0621313306** | **0.05434159731** |

Many thanks,

The Authors

---

### Author Response · Authors · 2025-12-03
**Summary of the Rebuttal Period**

**Final changelog for new AC**

Dear Reviewers and AC,

Many thanks for your careful and constructive reviews of our paper. We would like to briefly summarise the changes to our paper, given the AC reassignment and premature closure of the rebuttal phase.

The new manuscript now includes:

- **On the FNO comparison:**
  - An ablation study on four parameters, testing 144 points in hyperparameter space
  - Further exposition on the architecture and training details

- **On the DeepONet comparison:**
  - Further exposition on the architecture and training details

- **On the randomised tokenizer:**
  - An additional experiment in which we randomise the convolutional filter, instead of just average pooling
  - Exposition on the results of this experiment; it shows exactly the result we expect given our theory section

- **On the mathematical clarity of the paper:**
  - A more formal definition of the operator $G$
  - An argument demonstrating observability with probability 1 when the time evolution operator is perturbed slightly
  - A clarification of the $y(t, x), y(t)$ notation

- **On the rhetorical clarity of the paper:**
  - Significantly emphasise our contribution of observability as an explaining factor in world models
    - To this end, added a new experiment comparing the unmodified, unobservable wave equation with the modified observable counterpart
    - Introduced a new table clarifying which models and experiments are observable and which are not
  - Changed references from 'our model' to 'our approach'
  - Heavily clarified our novel contributions
  - Entirely rewritten the introduction to focus on operator learning, and to distinguish our work from the literature background
  - Edited the theorem statements to emphasise the observability requirement

The most common issue raised by the reviewers is the lack of rhetorical clarity, in particular with regard to our contributions. We believe, without putting words in their mouth, that this misunderstanding led to the criticisms of Reviewer 1aew. We hope we were beginning to convince them otherwise before the discussion was closed; see their comment for more details. Reviewer 36Bq raised similar concerns, but with some work was able to realise the significance of our method, and requested changes to the rhetorical presentation. We have tried hard during this rebuttal period to rectify these problems (point 5 above), and the results of these efforts can be found in the updated paper.


Many thanks,
The Authors

---

### Note · Authors · 2026-05-12

I have read and agree with the venue's withdrawal policy on behalf of myself and my co-authors.

---

### Meta-Review · Area_Chair_BwSF · 2025-12-17

**Summary:**

This paper targets what it calls "world models" by proposing an operator learning framework and a theory to analyze "probably approximately correct (PAC) learning". This approach, which shares similarities with GAN-based video generation, is compared to FNO and DeepOnet baselines, and is found to have a positive impact on learning tasks.

Despite interesting aspects, reviewers raised a series of fundamental concerns:
- about the theoretical contributions
- the novelty of the "world model" viewpoint
- the requirement for global observability
- the relatively weak baselines (no other "modern" architectures)
- a lack of clarity in the exposition was also mentioned several times

**Reviewer Concerns:**

The authors have responded with detailed posts that addressed concerns about rhetorical clarity, partially addressed “circularity” and strength of assumptions, misunderstandings about quantitative comparisons, and randomized tokenization.

However, the following concerns remain largely unaddressed: concerns about theoretical contributions, "world model" viewpoint novelty, global observability requirements, and the baselines.

**Reviewer Scores:**

In terms of ratings, I see the following outcome as very likely:
* 1aew - score of 2 preserved
* 36Bq - score of 6 potentially could have seen an increase to 8
* yVYL - kept score of 4
* DW7U - score of 4, has potential for a raise to 6

Overall, this puts the paper on the fence with an average of 5, which, for a renowned venue like ICLR is typically not enough for an accept. Given the fundamental concerns, I recommend a reject for this paper. I can recommend that the authors revisit and rephrase their world model approach, and improve their exposition and general clarity in a re-submission.

---

### Decision · Program_Chairs · 2026-01-26

Reject